# Female mice lacking *Ftx* lncRNA exhibit impaired X-chromosome inactivation and a microphthalmia-like phenotype

Yusuke Hosoi[1], Miki Soma[1], Hirosuke Shiura[1,3,6], Takashi Sado [4], Hidetoshi Hasuwa[5,7], Kuniya Abe[3], Takashi Kohda [1,6], Fumitoshi Ishino[1] & Shin Kobayashi [1,2]

X-chromosome inactivation (XCI) is an essential epigenetic process in female mammalian development. Although cell-based studies suggest the potential importance of the *Ftx* long non-protein-coding RNA (lncRNA) in XCI, its physiological roles in vivo remain unclear. Here we show that targeted deletion of X-linked mouse *Ftx* lncRNA causes eye abnormalities resembling human microphthalmia in a subset of females but rarely in males. This inheritance pattern cannot be explained by X-linked dominant or recessive inheritance, where males typically show a more severe phenotype than females. In *Ftx*-deficient mice, some X-linked genes remain active on the inactive X, suggesting that defects in random XCI in somatic cells cause a substantially female-specific phenotype. The expression level of *Xist*, a master regulator of XCI, is diminished in females homozygous or heterozygous for *Ftx* deficiency. We propose that loss-of-*Ftx* lncRNA abolishes gene silencing on the inactive X chromosome, leading to a female microphthalmia-like phenotype.

[1] Department of Epigenetics, Medical Research Institute, Tokyo Medical and Dental University (TMDU), 1-5-45 Yushima, Bunkyo-ku, Tokyo 113-8510, Japan. [2] Molecular Profiling Research Center for Drug Discovery, National Institute of Advanced Industrial Science and Technology, 2-4-7 Aomi, Koutou-ku, Tokyo 135-0064, Japan. [3] Technology and Development Team for Mammalian Genome Dynamics, RIKEN BioResource Research Center, 3-1-1 Koyadai, Tsukuba, Ibaraki 305-0074, Japan. [4] Department of Bioscience, Graduate School of Agriculture, Kindai University, 3327-204, Nakamachi, Nara 631-8505, Japan. [5] Research Institute for Microbial Diseases, Osaka University, Yamadaoka 3-1, Suita, Osaka 565-0871, Japan. [6] Faculty of Life and Environmental Sciences, University of Yamanashi, 4-4-37 Takeda, Kofu, Yamanashi 400-8510, Japan. [7] Present address: Department of Molecular Biology, Keio University School of Medicine, Tokyo 160-8582, Japan. Correspondence and requests for materials should be addressed to S.K. (email: kobayashi.shin@aist.go.jp)

X-chromosome inactivation (XCI), a unique epigenetic gene-regulating process in female mammals, equalizes X-linked gene expression between the sexes by inactivating one of the two X chromosomes in female eutherian mammals. XCI is essential for normal mammalian development because a double dose of X-linked gene products causes embryonic death[1]. One of the critical regulators of XCI is the *Xist* long non-protein-coding RNA (lncRNA), which is expressed from the inactive X chromosome and acts as the trigger for chromosome-wide silencing[1,2]. So far, several mouse lines with XCI defects have been reported, such as *Xist-*, *Tsix-* and *Rnf12*-deficient mice[1,3–5]. These mutant female mice always show embryonic death in utero.

In addition to *Xist*, the involvement of some other lncRNAs has been proposed for silencing the entire X chromosome, e.g., *Jpx*, *Ftx*, *Linx*, *Tsix*, *Xist-AR* and *Xite* lncRNAs[6]. One such candidate lncRNA is *Ftx*, which we and others have reported as a possible regulator of XCI[7,8]. We are particularly interested in *Ftx* for the following reasons. First, *Ftx* starts its expression predominantly in female preimplantation embryos when XCI first occurs. Furthermore, *Ftx* is imprinted so that it is expressed only from the paternally derived X chromosome ($X^P$), like *Xist* at these stages. In mice, there are two forms of XCI: imprinted and random. Imprinted XCI is initiated in preimplantation embryos and involves silencing of the $X^P$ in all cells, whereas random XCI of either the $X^P$ or $X^m$ (maternally derived X) chromosomes is initiated in the epiblast lineage after implantation[9–13]. Thus, *Ftx* is expressed from an inactive X at preimplantation stages. Second, *Ftx* is located within a cis-acting regulatory locus on the X chromosome, referred to as the "X-inactivation centre (Xic)", which is essential as a genomic element for XCI[14,15]. Third, apart from our study, Chureau et al. proposed that the *Ftx* lncRNA is an upregulator of *Xist*, as deletion of the *Ftx* promoter led to decreased *Xist* expression in male embryonic stem (ES) cells[7]. However, its role in female XCI remains to be elucidated.

To investigate its role in XCI, we generated *Ftx*-deficient mice[16]. Based on *Ftx* expression patterns at preimplantation embryo stages, we previously analysed its function in imprinted XCI. When the targeted allele was transmitted paternally, the *Ftx* expression was lost in female embryos heterozygous for the *Ftx* mutation. However, as far as we examined, *Ftx* deficiency had no apparent effects on imprinted XCI at preimplantation stages[16].

In this study, we examine the function of *Ftx* in vivo after implantation, particularly its potential importance in random XCI. We find that female mice homozygous for *Ftx* deficiency are viable and fertile, contrasting with the early embryonic death commonly observed in XCI mutants. Intriguingly, a subset of *Ftx*-deficient mice displays an obvious defect in eye development reminiscent of microphthalmia in humans. Its inheritance pattern is unusual in that it occurs substantially only in female but not in male mice. Female mice deficient in *Ftx* fail to silence genes properly on the inactive X chromosome, with the expression level of *Xist* being diminished. Such misexpressed genes are not clustered but scattered across the X chromosome, indicating that *Ftx* could be involved in appropriate regulation of X chromosome inactivation. We propose that a functional loss-of-*Ftx* lncRNA abolishes the proper silencing of genes on the inactive X chromosome, which can directly or indirectly cause a microphthalmia-like phenotype in a substantially female-specific manner.

## Results

**Eye defects in *Ftx*-deficient mice with unusual inheritance**. We reported previously that *Ftx* was imprinted and expressed predominantly in female blastocysts at the preimplantation stage[8] (Supplementary Fig. 1a). Here we examined *Ftx* expression patterns in post implantation embryos as well as adult tissues (Supplementary Fig. 1a–c). *Ftx* was expressed in both males and females, indicating that *Ftx* had lost its imprinted status (Supplementary Fig. 1a, b). To examine the allelic expression pattern of *Ftx* in female embryos, we took advantage of DNA polymorphisms in F1 female mouse embryonic fibroblasts (MEFs) heterozygous for a *Xist* null mutation ($Xist^{-/+}$), in which the wild-type (WT) X chromosome derived from *Mus mus molossinus* ($X^{+JF1}$) is inactivated selectively because the mutated *Xist*-deficient X of *M. m. musculus* origin ($X^{-B6}$) never undergoes XCI. As shown in Supplementary Fig. 1d, *Ftx* was expressed from both the active and inactive X chromosomes, indicating that it escaped random XCI. To examine the effects of functional deficiency of *Ftx* in female offspring, we crossed heterozygous female with hemizygous male mice. All the mating experiments produced offspring at the expected Mendelian ratios, and no apparent abnormalities were observed in litter sizes (Supplementary Table 1). Homozygous female mice, in which *Ftx* expression was completely lost (Supplementary Fig. 1e), were viable and fertile. However, to our surprise, some of the *Ftx*-deficient mice showed abnormalities in eye development (Fig. 1a). When we removed the eyes surgically, we found they were very small or almost absent.

The pedigree chart from a male chimera showed that the phenotype was observed in female mice but not in hemizygous ($Ftx^-$/Y) male mice, indicating that the inheritance pattern matched neither X-linked dominant nor recessive mutations (Fig. 1b, c; individuals showing eye abnormalities are marked with red). This is unusual for the X-linked phenotype, because both X-linked dominant and recessive mutations usually show more severe phenotypes in hemizygous male than in heterozygous female mice. In addition, some female mice heterozygous for the *Ftx* mutation also showed this abnormal eye phenotype regardless of the parental origin of the targeted allele, indicating that the inheritance pattern did not fit with a model of genomic imprinting (Fig. 1b). The pedigree shown in Fig. 1c also shows that daughters of affected mothers were not necessarily affected. Thus, the *Ftx* mutation presents a very unusual inheritance pattern where the abnormal eye phenotype appeared stochastically only in a subset of homozygous females ($Ftx^{-/-}$), with 5 (Fig. 1b) and 11 (Fig. 1c) mice affected. Three heterozygous female ($Ftx^{+/-}$, $Ftx^{-/+}$) mice were affected (Fig. 1b) but no hemizygous male mice were affected (Fig. 1b, c).

**No phenotype in mice deficient in both *miRNA374* and *miRNA421*.** Because intron 12 of the *Ftx* lncRNA harbours two microRNAs—*miRNA374* and *miRNA421*—our scheme targeting *Ftx* abolished not only *Ftx* lncRNA but also *miRNA374/421* expressions on the targeted X chromosome (Fig. 2a; middle panel)[16]. To clarify whether the observed eye phenotype was caused by a loss-of-*Ftx* lncRNA or miRNAs, we generated double knockout (DKO) mice deficient for both *miRNA374* and *miRNA421* (*miRNA374/421* DKO) (Fig. 2a lower panel; b–f). Female mice heterozygous for *miRNA374/421* DKO were crossed with male mice hemizygous for this DKO (Supplementary Table 2). We confirmed that homozygous DKO lacked the expression of both *miRNA374* and *miRNA421* (Fig. 2f), and the deletion did not show obvious effects on the expression of the host transcript: *Ftx* lncRNA (Supplementary Fig. 1f). Then, we examined the incidence of gross eye abnormalities at weaning. None of the mice deficient in *miRNA374/421* showed overt abnormalities in the eye (Fig. 2g, *miRNA374/421* DKO), while eye abnormalities were observed in 7.1% and 17% of the female mice heterozygous and homozygous for the *Ftx* mutation, respectively (Fig. 2g, *Ftx* KO #64). A similar result was obtained in another

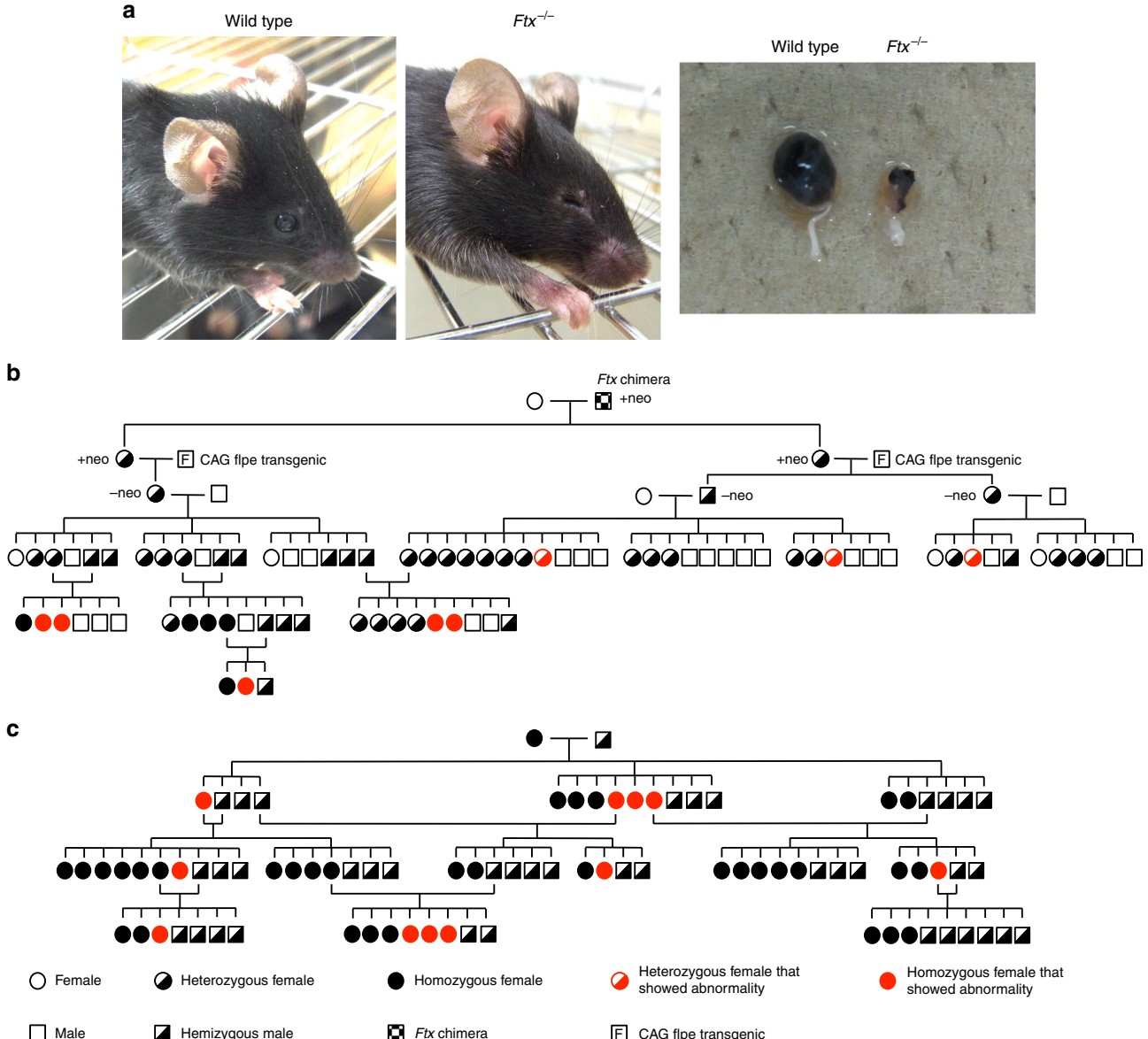

**Fig. 1** Eye abnormalities in *Ftx*-deficient mice showed an unusual X-linked inheritance pattern. **a** Side view of wild-type (WT) and $Ftx^{-/-}$ adult female mice. The right most panel shows the surgically removed eyes from WT and $Ftx^{-/-}$ adult female mice respectively. **b** A pedigree obtained from an analysis of an *Ftx*-deficient chimera. **c** A pedigree is shown for the outcomes of crossing homozygous female with hemizygous male mice

*Ftx*-deficient line generated using the ES cell line #77. These findings demonstrated that the eye abnormalities were caused by the elimination of *Ftx* but not by *miRNA374/421*.

**Human microphthalmia-like phenotype in *Ftx*-deficient mice.** To analyse the eye abnormalities in more detail, we isolated $Ftx^{-/-}$ embryos at different stages of development. The abnormalities had already become evident in some foetuses by embryonic day (E) 15.5 and E13.5 (Fig. 3a). As these eye abnormalities were variable, we classified the severity of the phenotypes into three categories: type 1, normal; type 2, mild; and type 3, severe (see details in Fig. 3a legend). As was the case at the weaning stage, these eye abnormalities were common in the embryos from $Ftx^{-/-}$ mutants (Fig. 3b; types 2 and 3) but not in those for *miRNA374/421* DKO mutants (Fig. 3b; *miRNA374/421*-DKO$^{-/-}$). Male $Ftx^{-}$/Y embryos almost never showed abnormalities (Fig. 3b, type 2 $Ftx^{-/-}$ vs. $Ftx^{-}$/Y, $P < .01$; type 3 $Ftx^{-/-}$ vs. $Ftx^{-}$/Y, $P < .05$).

For further characterization of the abnormalities, we performed histological analysis of the eyes isolated from each class of the mutants (Fig. 3c). Morphology of the sectioned eye in type 1 $Ftx^{-/-}$ embryos was indistinguishable from that in WT embryos, as far as we observed. In type 2 mutants, several malformations were observed in the anterior segment of the eye, although the layers of the neural retina appeared to be organized normally. Combined with this histological observation, large defects in the ocular tissues, typically the absence of tissues, or gaps in the iris, were commonly observed in type 2 eyes (Fig. 3a), indicating that these defects closely resemble human ocular defects known as "coloboma of the iris". Colobomas result from an aberrant closure of the embryonic fissure of the optic cup, and appear as a congenital gap in the iris[17]. This condition is frequently associated with small eyes (microphthalmia) or their complete absence (anophthalmia) as part of an interrelated spectrum of developmental eye anomalies[18] (OMIM #120200). Type 3 mutants showed severe histological abnormalities in eyes wherein

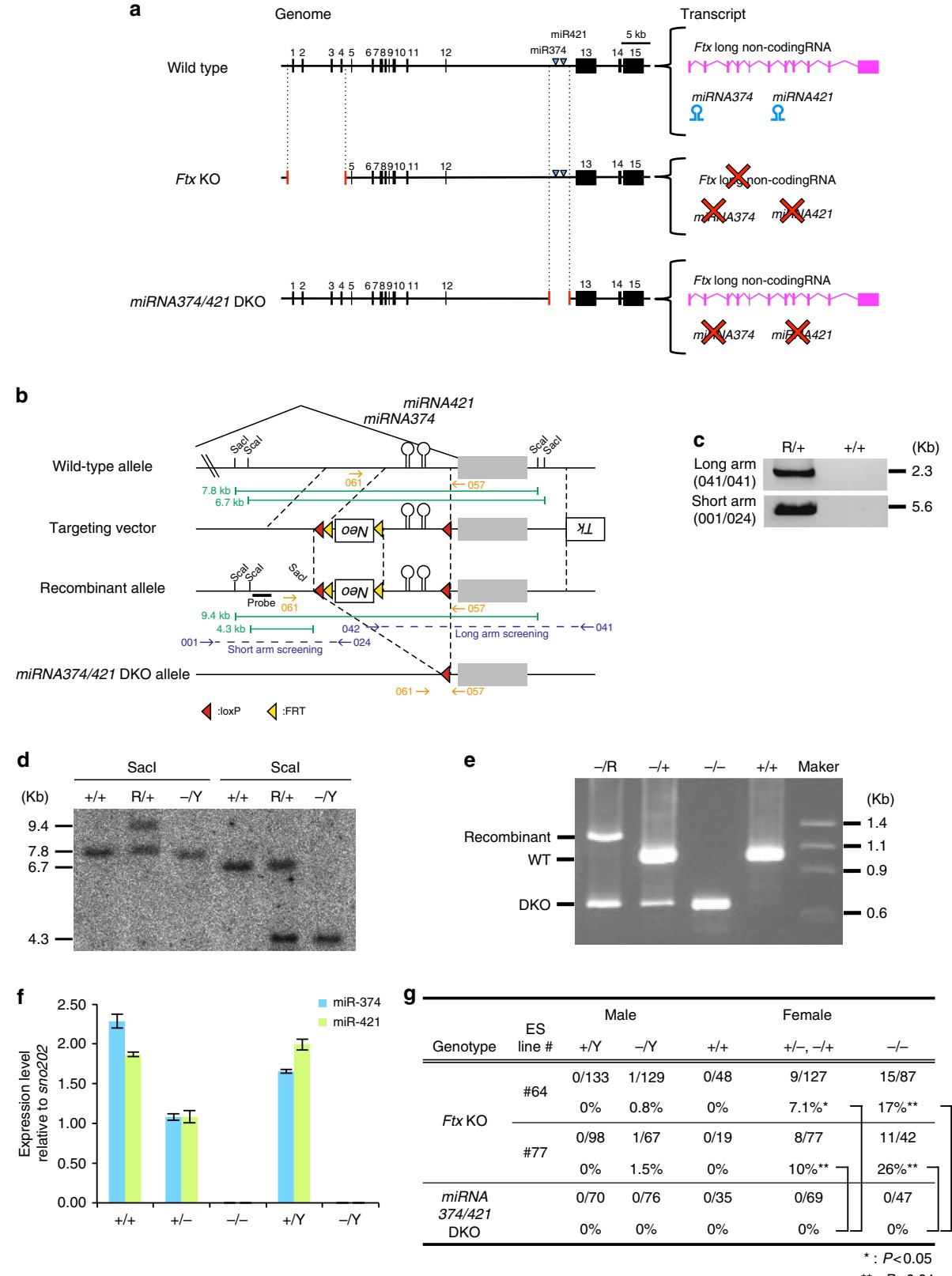

a lens was not formed, but retinal epithelial tissue was formed inside the body (Fig. 3c). It resembled the human phenotype of severe microphthalmia or anophthalmia[19]. Furthermore, measuring the size of the eyes using haematoxylin and eosin (HE) staining of sections confirmed that types 2 and 3 eyes were significantly smaller than those in WT embryos (Fig. 3d). Taking

these findings into consideration, we conclude that eye abnormalities of $Ftx^{-/-}$ mice appear to closely resemble human microphthalmia.

**$Ftx^{-/-}$ mice exhibited partial failure of XCI.** Why were only female offspring affected adversely by functional loss of $Ftx$ in

**Fig. 2** Mice doubly deficient for *miRNA374* and *miRNA421* showed no particular phenotype. **a** Map of the WT, *Ftx*-deficient (*Ftx* KO) allele, and *miRNA374/ 421*-doubly deficient (*miRNA374/421*-DKO) allele. The expression of both lncRNA *Ftx* and two microRNAs—*miRNA374* and *miRNA421*—were lost in the *Ftx*-targeted allele. **b** Scheme for generating the miRNA374/421-DKO allele. The neomycin-resistance gene (*Neo*) was used as a positive selection marker. Yellow and red triangles indicate FRT and loxP sites, respectively. **c** Homologous recombination was confirmed by PCR. The PCR primers (001, 024, 041 and 042) used for genotyping are depicted in **b**. R, recombinant allele. **d** Homologous recombination was confirmed by southern blotting. The probe position is denoted as a thick black rectangle in **b**. Genomic DNA was digested with SacI (left) and ScaI (right). **e** Cre-loxP-mediated recombination was confirmed by PCR. The PCR primers (061 and 057) used for genotyping are depicted in **b**. R, recombinant allele. Marker lane: size marker DNA, ΦX174/ Hae III digest. **f** The expression levels of miRNA374 and miRNA421 were quantified by RT–qPCR in adult mouse kidneys ($n = 3$) of WT females (+/+), heterozygous females (+/−), homozygous females (−/−), WT males (+/Y) and hemizygous males (−/Y). Error bars, s.d. **g** The frequency of gross eye abnormalities in *Ftx*-deficient and *miRNA374/421*-DKO mice at weaning. The abnormalities were only observed in *Ftx*-deficient (−/+, +/− and −/−) mice but not in *miRNA374/421*-DKO (−/+, +/− and −/−) mice ($\chi^2$-tests; *$P < .05$; **$P < .01$)

their eyes? Given that the phenotype was almost entirely specific to female mice and that *Ftx* is one of the non-coding Xic genes proposed to be involved in the regulation of *Xist*, it was of interest to determine XCI status in female mice homozygous for the *Ftx* mutation. We compared gene expression patterns of WT and type 1 *Ftx*$^{-/-}$ eyes (at E13.5) using a DNA microarray (Fig. 4a). The results demonstrated that altered expression was exhibited by a large number of genes in the mutant females, but by far fewer genes in the hemizygous mutant males. Altered expression patterns (*Ftx*$^{-/-}$ vs. WT females and *Ftx*$^-$/Y vs. WT males) are shown as upregulated and downregulated genes in Fig. 4b (upper panels show autosomal genes; lower panels show X-linked genes) and as a scatter plot in Fig. S2. Many X-linked genes were markedly altered in their expression in female mice, but far fewer were altered in male mice.

To analyse the distribution of these affected genes along the X chromosome, the fold changes (FCs) of each X-linked gene in the mutant female mice (*Ftx*$^{-/-}$/WT) were plotted according to a physical map of the X chromosome. As shown in Fig. 4c, several upregulated genes were detected throughout the entire X chromosome (Fig. 4c, upper panel: 93 genes marked as red dots, *Ftx*$^{-/-}$ vs. WT females; FC > 1.2, $P < .05$). We consider these genes to be candidates for those "escaping" XCI in a manner specific to female mice homozygous for the *Ftx* mutation. Comparisons between WT female and WT male mice (Fig. 4c, lower panel) support this hypothesis: first, most genes were classified within the range of $0.8 < FC < 1.2$; second, five known gene "escapees" were included in those genes with FC > 1.2, indicating that the microarray analysis appeared to be sensitive enough to detect these anomalies in XCI. Taking these findings together, these results suggest that some of the X-linked genes failed to be inactivated and were overexpressed in *Ftx*$^{-/-}$ female mice.

Figure 4d shows a list of genes differentially expressed between *Ftx*$^{-/-}$ and WT female mice. Excluding genes containing repetitive sequences, five out of 18 listed genes showed more than a 1.4-fold (140%) increase in their expression levels. Quantitative reverse transcription polymerase chain reaction (RT–qPCR) was carried out to re-examine their expression levels. Of these, three genes—*Tmem29*, *Mecp2* and *Ogt*—were significantly expressed at higher levels in *Ftx*$^{-/-}$ female mice than in WT mice (Fig. 4e). No significant increases were detected in *Gab3* or *Slc16a2*, presumably because of their low expression levels and/or non-ideal probe design.

We also studied whether XCI was actually impaired in *Ftx*$^{-/-}$ females. RNA fluorescent in situ hybridization (FISH) was carried out for nascent RNA of the three candidate escapee genes: *Tmem29*, *Mecp2* and *Ogt*, as well as *Xist* as a marker of the inactive X chromosome. As shown in Fig. 5a, hybridization signals for their nascent RNA were detected in the *Xist* cloud in a subset of cells of the eye (but a significant portion of them) in *Ftx*$^{-/-}$ female mice but not in cells from the WT mice, indicating

misexpression of these genes from the inactive X chromosome in *Ftx*$^{-/-}$ female mice (Fig. 5b, shown as dark and light pink in the graphs; see details in the figure legend; control male samples using RNA–FISH are shown in Supplementary Fig. 3a). Their ectopic expression from the inactive X chromosome was evident in the mutant females. By contrast, the non-upregulated *Pgk1* gene barely showed misexpression from the inactive X chromosome, supporting the idea that not all X-linked genes were affected (Supplementary Fig. 3b). Although there was no apparent difference in the numbers of *Xist* cloud-positive cells between WT and *Ftx*$^{-/-}$ cells (Fig. 5b), the size of the *Xist* cloud in *Ftx*$^{-/-}$ was slightly—but significantly—smaller than in WT cells (Supplementary Fig. 4; WT vs. *Ftx*$^{-/-}$ eye).

Furthermore, taking advantage of a single nucleotide polymorphism found in two X-linked genes between *Mus musculus musculus* (C57BL/6N) and *Mus musculus molossinus* (JF1) (Supplementary Figs 5, 6a and 7a), we also carried out allelic expression analysis to confirm the impairment of random XCI in heterozygous *Ftx*$^{+/-}$ eyes (Supplementary Figs 5, 6 and 7). In these experiments, we obtained embryos by crossing B6 background X$^{GFP}$ WT mice and JF1 backcrossed *Ftx*$^-$/Y mice. The enhanced green fluorescent protein (eGFP)-positive cells were collected from E13.5 eyes using fluorescence activated cell sorting (FACS), and the expression levels of two X-linked genes, *Mecp2* and *Ogt*, were examined using RT–qPCR and restriction length polymorphism (RFLP) analysis. In contrast to the 100% pure B6 homozygous *Ftx*$^{-/-}$ cells, we could not find statistically significant upregulation of *Mecp2* or *Ogt* in F1 hybrid *Ftx*$^{+/-}$ heterozygous mutants (Supplementary Figs 6b–7b). However, RFLP analysis showed ectopic expression from the inactive X chromosome in the *Ftx* mutants, indicating that *Ftx* deletion could well affect XCI and *Ftx* could regulate XCI in cis (Supplementary Figs 6c–7c: the position of the misexpressed allele is indicated with a red arrowhead). All these results support our idea that some of the X-linked genes were misexpressed from the inactive X chromosome, and random XCI was partially compromised in *Ftx*$^{-/-}$ and *Ftx*$^{+/-}$ females.

Because, *Ftx* was ubiquitously expressed in all the foetal stages and the adult tissues we examined (Supplementary Fig. 1a–c), we investigated whether XCI was also compromised in other tissues in the *Ftx*$^{-/-}$ female mice. RNA–FISH revealed that misexpressions of *Tmem29* and *Mecp2* were detected in the brain and kidney to a similar extent as in the eye (Fig. 5c), and the *Xist* cloud area in mutants was also smaller than that in WT embryos (Supplementary Fig. 4, WT vs. *Ftx*$^{-/-}$ brain, WT vs. *Ftx*$^{-/-}$ kidney). This suggests that the importance of *Ftx* in random XCI is not confined to the eye but extends to multiple tissues in the whole body.

Furthermore, to make the status of this XCI impairment clearer, we carried out two more experiments. First, we performed RNA–FISH of *Ftx*$^{-/-}$ embryos at E6.5 when random XCI is

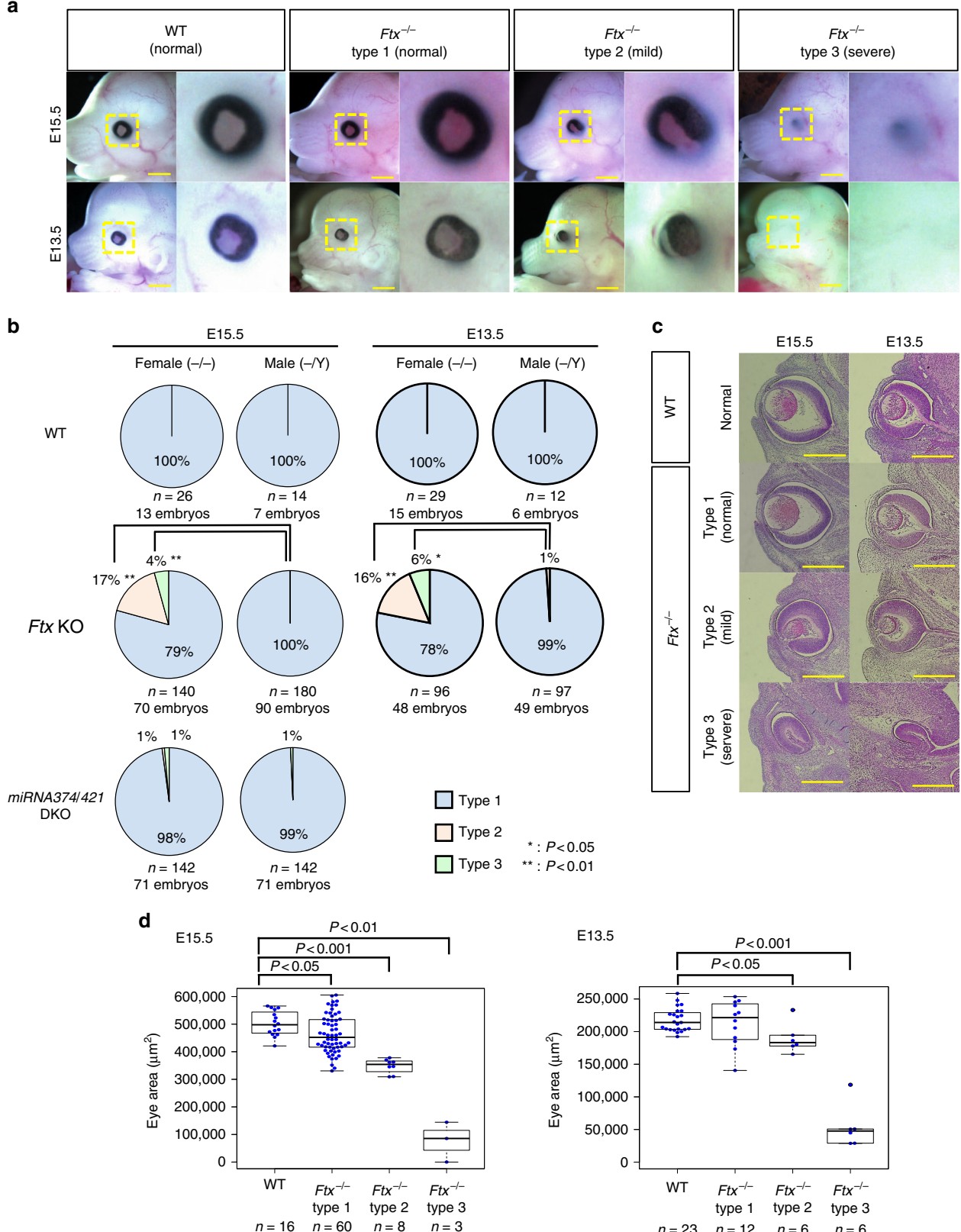

initiated or soon after (Supplementary Fig. 8a, b). Misexpression of *Mecp2* was observed in the *Ftx*⁻/⁻ embryos, suggesting the involvement of *Ftx* in the initiation step of random XCI. Comparing the percentage of cells showing misexpression in E6.5 (2.7%) (Supplementary Fig. 8b) with that in E13.5 (eye 9.7%; brain 13.6%; kidney 7.1%; Fig. 5b, c), it is possible that *Ftx*

functions in the initiation and maintenance phases of random XCI. Second, to examine whether random XCI was affected in *Ftx*⁻/⁺ embryos and resulted in skewed expression in one of the X chromosomes, we carried out RFLP analysis of six X-linked genes, five non-upregulated genes and *Xist*, in (B6 × JF1) F1 hybrid embryos at E9.5 (Supplementary Fig. 9). The results for six

**Fig. 3** The eye abnormalities of $Ftx^{-/-}$ mice closely resemble human microphthalmia. **a** Classification of phenotypic variation in the $Ftx^{-/-}$ eyes. The yellow boxed areas are shown enlarged in the right panels. Abnormalities were classified into three types by their appearance: type 1 (normal) eyes were similar to WT eyes; in type 2 (mild) there was absent tissue or a gap in the iris; and in type 3 (severe) there was no eye visible on the face. Scale bar = 1 mm. **b** The frequency of abnormalities in the WT, $Ftx$-deficient ($^{-/-}$ or $^{-}$/Y) and $miRNA374/421$-DKO embryos ($^{-/-}$ or $^{-}$/Y). Differences between $Ftx$-deficient male and female mice were statistically significant for type 2 and type 3 abnormalities ($\chi^2$-tests; *$P < .05$, **$P < .01$). **c** HE-stained sections of WT and $Ftx^{-/-}$ eyes at embryonic day (E) 15.5 and E13.5. Scale bar = 1 mm. **d** Small ocular size in the $Ftx^{-/-}$ mice at E15.5 and E13.5. A subset of WT and KO eyes classified by their appearance in Fig. 3b was analysed. The analysed number of each type is shown at the bottom of the column. The box represents the median, 25th and 75th percentiles; whiskers indicate the maximum and minimum values. The sizes of eyes in HE-stained sections were measured using Image J software. The mean eye size was significantly different between the $Ftx^{-/-}$ (type2 and 3) and WT embryos (two-tailed, unequal variance $t$-test)

X-linked genes showed that deleting $Ftx$ on one X chromosome did not skew XCI completely toward the WT X chromosome in female somatic cells. Furthermore, $Ftx$ targeting the B6 allele tended to show lower $Xist$ expression than in WT cells, suggesting that $Ftx$ regulates $Xist$ expression in cis.

**Loss of $Ftx$ compromised $Xist$ expression**. To address mechanistic aspects of the loss of $Ftx$ in XCI failure in $Ftx^{-/-}$ female mice, we examined its effect on the expression of $Xist$. Microarray data showed that the $Xist$ expression level tended to be lower in $Ftx^{-/-}$ female mice (11 eyes) than in WT female mice (three eyes; Fig. 6a). The RT–qPCR analysis revealed that the expression level of $Xist$ was significantly diminished even in type 1 $Ftx^{-/-}$ eyes, which were morphologically normal (Fig. 6b, eye, WT females vs. $Ftx^{-/-}$, $P < .01$). In addition to $Ftx^{-/-}$ eyes, we quantified $Xist$ expression in $Ftx^{+/-}$ eyes and compared it with that in WT eyes. With decreasing $Ftx$ expression levels in mutant eyes, the diminished $Xist$ expression was observed in $Ftx^{+/-}$ as well as in $Ftx^{-/-}$ embryos (Supplementary Fig. 10). We also examined $Xist$ expression levels in organs other than the eye and found that it tended to be diminished in several tissues (Fig. 6b, brain, kidney and liver). These data indicate a critical role for $Ftx$ in the proper regulation of $Xist$ in vivo.

We next addressed whether variation in the severity of the eye phenotype was associated with partial defects of random XCI. First, we examined whether the percentage of cells showing misexpression from Xi increased in type 2 eyes, and carried out RNA–FISH for $Tmem29$, $Mecp2$ and $Ogt$ (Supplementary Fig. 3c). However, the percentage of cells showing misexpression from Xi did not increase in type 2 eyes. Thus, we could exclude this possibility. Therefore, we investigated the relationship between the number of X-linked genes misexpressed from the inactive X chromosome (FC > 1.2) and the expression level of $Xist$ in $Ftx^{-/-}$ eyes. According to the microarray data, there was a negative correlation between the level of reduction in $Xist$ expression and the number of upregulated X-linked genes in $Ftx^{-/-}$ female mice (Fig. 6c, left panel). A similar negative correlation was also found between the expression levels of $Xist$ and the numbers of upregulated and downregulated autosomal genes in $Ftx^{-/-}$ mutant mice (Fig. 6c, middle and right panels). These results suggest that the lower the $Xist$ expression level, the more X-linked genes showed misexpression from the inactive X chromosome. Moreover, we found that the $Xist$ expression levels in the brain were more severely diminished in $Ftx^{-/-}$ type 2 mutants than in type 1 mutants, suggesting that the phenotypic severity of eye defects was correlated with the variability in diminished $Xist$ expression in tissues other than eyes (Fig. 6d). These data indicate that $Ftx^{-/-}$ could have a role in the proper regulation of $Xist$ in vivo, and that diminished $Xist$ expression might correlate with the variability of the impaired XCI and the severity of the resulting variable phenotype.

To further investigate the relationship between a reduction in $Xist$ expression levels and the eye phenotype, we carried out Gene Ontology analysis. X-linked overexpressed genes (FC > 1.2, $t$-test

$P < .05$) and autosomal upregulated genes did not show any significant enrichment of genetic pathways involved in eye development (Supplementary Table 3, $P < .05$; Supplementary Table 4, $P < .01$; Bonferroni adjusted). In contrast, autosomal downregulated genes were significantly enriched in genetic pathways involved in eye development (Fig. 6e, Supplementary Table 5, $P < .01$; Bonferroni adjusted). Indeed, 12 genes associated with eye development were decreased even in the type 1 mutant eyes, and their expressions were positively correlated with $Xist$ expression levels (Supplementary Fig. 11a). Moreover, it has been reported that genetic mutations in these 12 genes can cause microphthalmia (Supplementary Fig. 11b). Because $Ftx$ deficiency has a small effect on genome-wide gene expression in male mice, we propose that functional loss of $Ftx$ causes a failure in the proper silencing of genes on the inactive X chromosome, which directly or indirectly can lead to downregulation of genes involved in eye development and eventually to microphthalmia in a subset of females homozygous or heterozygous for the $Ftx$ mutation (Supplementary Fig. 12).

**Discussion**

We demonstrated here that a loss-of-$Ftx$ lncRNA resulted in misexpression of genes on the inactive X chromosome in the eye and other tissues with a concomitant reduction in $Xist$ expression, supporting the idea that $Ftx$ could have a critical role in the proper regulation of random XCI and $Xist$ expression in vivo. Given that microphthalmia occurred only in some of the female mice (17–26% of homozygous and 7.1–10% of heterozygous mutants) and barely at all in male mice, it is highly probable that a partial defect in XCI somehow caused these eye abnormalities. Analysis of the transcriptomes of eyes isolated from $Ftx$-deficient mice revealed that the female mice showed more severe genome-wide alterations in gene expression than did male mice. Although, we could not rule out possible roles in male mice, our data showed that $Ftx$ has a more significant function in females than in males. The few changes in gene expression in male mice suggest that $Ftx$ regulates autosomal gene expression, not directly in a trans-acting manner, but indirectly through misexpression of genes on the inactive X chromosome. Most of the affected alterations in autosomal gene expression in female mice might result through complex gene networks connecting autosomal genes and X-linked genes. In other words, we presume that the genome-wide changes in gene expression in female mice are secondary effects arising from the overexpression of some X-linked genes caused by partial XCI failure and that these changes can eventually lead to microphthalmia.

So far, there have only been two reports of mutant mice with XCI that show phenotypes other than embryonic death. One exception is that epiblast lineage-specific $Xist$ conditional knockout mice showed a range of severity; while some were born, almost all the mutants died before weaning had finished[20]. Another exception is the haematopoietic lineage-specific $Xist$ deletion resulting in loss of XCI and the development of blood cancer[21]. Both exceptions involved tissue-specific deletions of $Xist$

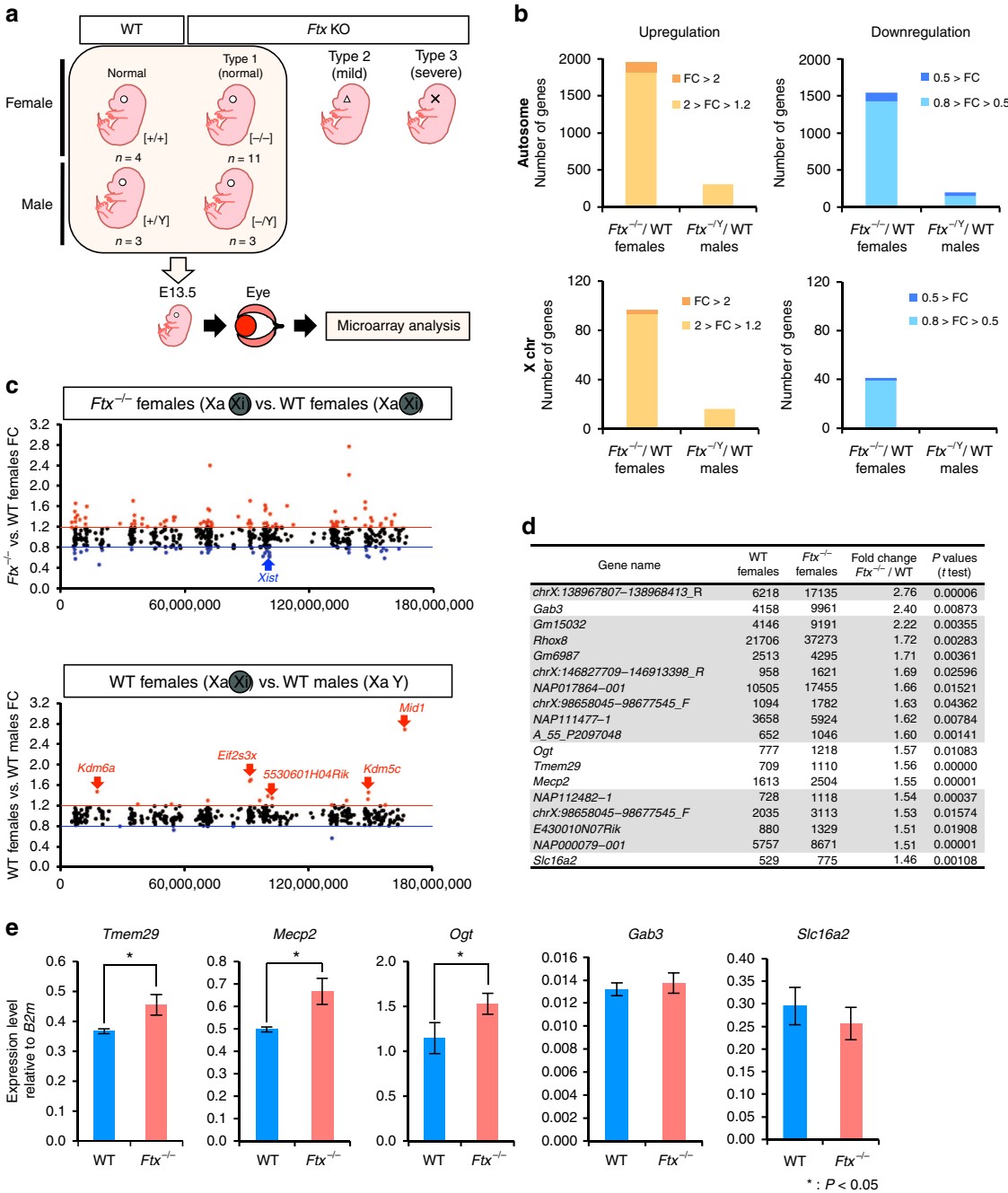

**Fig. 4** Analysis of the transcriptomes of eyes from *Ftx*⁻/⁻ mice. **a** Scheme for microarray analysis. WT female eyes ($n = 4$), WT male eyes ($n = 3$), *Ftx*⁻/⁻ female eyes (type 1, $n = 11$) and *Ftx*⁻/Y male eyes (type 1, $n = 3$) were used for the analysis. All eyes were removed from E13.5 embryos and were of normal appearance. **b** Numbers of genes whose expression level altered between WT and *Ftx*-deficient mice. Comparing between WT and *Ftx*⁻/⁻ female mice or between WT and *Ftx*⁻/Y male mice, the numbers of upregulated genes (fold change, FC, >1.2, $P < 0.05$) and downregulated genes (FC < 0.8, $P < 0.05$) are shown respectively (upper, autosomal genes; lower, X-linked genes). A two-tailed, unequal variance *t*-test ($P < .05$, WT $n = 3$ vs. KO $n = 11$) was used for statistical analysis to calculate FC values. **c** Physical location map of X-linked genes on the X chromosome. FC values of the *Ftx*⁻/⁻ females vs. WT females are plotted in the upper panel, and FC values of WT females vs. WT males are plotted in the lower panel. Red dots show upregulated genes with FC >1.2, and blue dots show downregulated genes with FC < 0.8. Red arrows show genes known to escape from XCI. Xa, active X chromosome; Xi, inactive X chromosome marked with shaded circles. *Ftx* was not indicated in this panel because of its low signal intensity (signal intensity <500). **d** List of upregulated X-linked genes sorted in descending order of their FC values. The grey rows show genes containing repetitive sequences. Lists of all the X-linked genes and autosomal genes (signal intensity >500) are provided in Supplementary Data 1 and 2, respectively. **e** RT–qPCR analysis of *Tmem29, Mecp2, Ogt, Gab3* and *Slc16a2* in WT and *Ftx*⁻/⁻ eyes ($n = 3$, E13.5). Significance levels of the mean differences between WT and *Ftx*⁻/⁻ eyes were calculated using two-tailed, unequal variance *t*-tests. Error bars, s.d.; *B2m*, beta-2 microglobulin

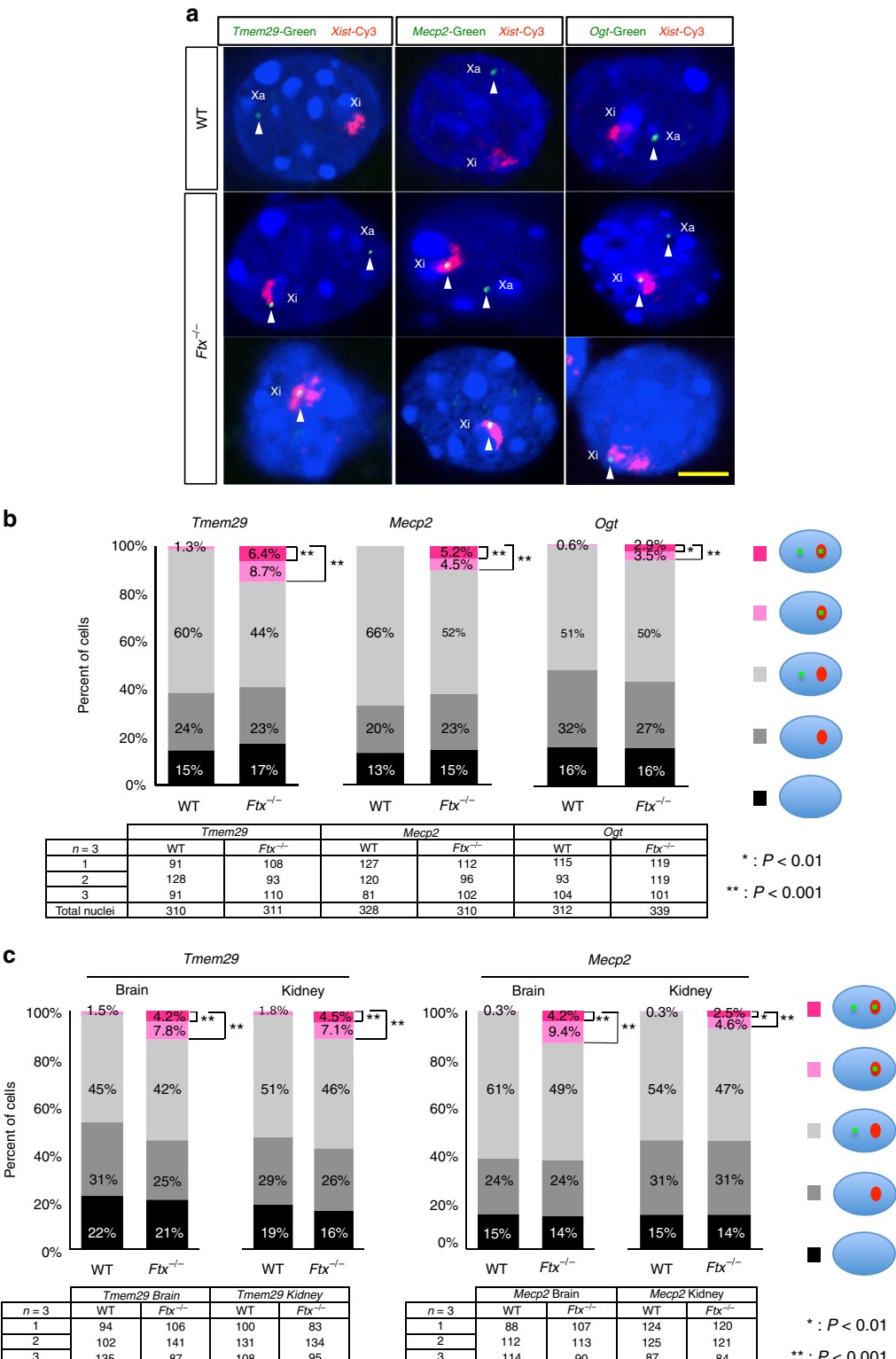

in mutant mice. In contrast, our *Ftx*-deficient mice were conventionally gene-targeted but were born alive, reached adulthood and were fertile. As far as we examined, there was no detectable phenotype other than eye abnormalities. Our data showed that the failure of dosage compensation occurred across multiple tissues in *Ftx*-deficient mice, but that failure was limited to a subset of X-linked genes. We suggest that this partial XCI failure might

help to explain why the *Ftx*-deficient mice did not die and were thus able to develop to term.

Although, the eye abnormalities showed varying degrees of severity, the stochastic nature of this phenotype cannot be explained simply by differences in the genetic background because *Ftx*-deficient mice were created and maintained on a pure C57/BL6 background (see Methods). By contrast, the diminished

**Fig. 5** $Ftx^{-/-}$ mice showed partial failure of XCI. **a** Representative images of RNA–FISH analysis in WT and $Ftx^{-/-}$ female eyes at E13.5, using $Xist$ (in red) and each X-linked gene (in green) as probes. Xa, active X chromosome; Xi, inactive X chromosome. Scale bar = 5 mm. **b** Frequency of ectopic X-linked gene expression from inactive X chromosomes in eyes at E13.5. Approximately 100 nuclei in each eye were counted from three independent experiments and classified based on the patterns of $Xist$ clouds and nascent X-linked gene transcripts. Cells showing misexpression from the inactive X chromosome are shown as dark or light pink, indicating that XCI was partially compromised in the mutant female (the percentage of the cells labelled in dark pink show signals from the active and inactive X; the percentage of the cells labelled in light pink show signals from inactive X only). Statistically significant differences between WT and $Ftx^{-/-}$ mice were observed by Chi-squared tests; *$P < .01$, **$P < .001$). Total cell numbers analysed in each embryo are shown in the table. Nuclei with more than one $Xist$ cloud were not observed. **c** Frequency of ectopic X-linked gene expression from the inactive X chromosome in brain and kidney cell nuclei at E13.5. Approximately 100 nuclei were counted for each sample, and statistically significant differences were detected between WT and $Ftx^{-/-}$ cells using $\chi^2$-tests (*$P < .01$, **$P < .001$)

$Xist$ expression levels in $Ftx$ mutants might explain this phenotypic variation. Even though partial XCI failure and the diminishing $Xist$ expression were observed in type 1 $Ftx$-deficient mice—which are phenotypically indistinguishable from WT mice—we speculate that the $Ftx$ mutants with types 2 and 3 abnormal eyes might have had more severe failure in XCI, and lower $Xist$ expression levels. To support this hypothesis, $Xist$ in type 2 brains showed lower expression than that in type 1 and WT brains. The more $Xist$ expression was diminished, the greater the number of X-linked genes that were overexpressed. Reduced $Xist$ expression might result in a more severe phenotype, although the molecular mechanism by which $Ftx$ regulates the expression of $Xist$ remains to be investigated. The epigenetic instability of XCI may explain these variable phenotypes, which cannot be explained simply from conventional genetics.

Based on these findings, we propose that partial failure in XCI might also cause genetic diseases with female-specific or female-predominant inheritance in humans. As far as we know, there is no reported human family showing microphthalmia to be inherited in a female-specific manner. Although the symptoms are different, X-linked genetic diseases such as craniofrontonasal syndrome (OMIM 304110) and early infantile epileptic encephalopathy-9 (OMIM 300088), which show greater sensitivities in female than in male subjects, suggest that they might be caused by partial XCI failure. Because, the $Ftx$ lncRNA is conserved in humans, we suggest that mutational analysis of human $FTX$ homologues might help us to understand this unusual class of X-linked human genetic diseases, for which women show more severe phenotypes than do men.

While our manuscript was under revision, an independent study, Furlan et al., reported that $Ftx$ is required both in the upregulation of $Xist$ and control of X-inactivation in ES cells[22]. The in vitro data presented in Furlan et al., complements our in vivo analysis and, together, both studies clarify the importance of $Ftx$ in random XCI.

## Methods

**Animals**. All experimental animals were handled according to the guidelines of the Committee on the Use of Live Animals in Teaching and Research of Tokyo Medical and Dental University. All efforts were made to minimize suffering. The details of generating the $Ftx$-deficient mice were as described[16]. Importantly, gene targeting was carried out using ES cells (EGR-G101) established from C57BL/6N mice (Clea Inc., Japan), and $Ftx$-deficient mice were maintained on a 100% pure C57BL/6N mouse strain background. Eight- to sixteen-week-old mice were used in mating experiments.

**Generation of $miRNA374/421$ doubly deficient mice**. The targeting vector was constructed to insert the Neo cassette (flanked by $FRT$ sites) to delete the $miRNA374$ and $miRNA421$ sequences ($miRNA374/421$ DKO) using pNT1.1. A herpes simplex virus thymidine kinase gene ($Tk$) was introduced as a negative selection marker cassette at the 3′-end of the targeting vector. Primers used for the construction of $miRNA374/421$ DKO targeting vector are listed in Supplementary Table 6. The linearized NotI-digested targeting vectors were electroporated into ES cells derived from 129/SvJ mice. After confirming accurate homologous recombination, cells from the targeted cell line were injected into C57BL/6N blastocysts,

and male chimeric mice were generated. After deleting the Neo cassette by crossing with CAG-Cre transgenic mice (B6.Cg-TG(CAG-Cre)), the progenies were back-crossed to C57BL/6N mice for over seven generations. Primers for confirming homologous recombination are listed in Supplementary Table 6. The $miRNA374/421$ DKO mouse line and the $Ftx$-deficient mouse line have been submitted to RIKEN BRC (http://www.brc.riken.jp/inf/en/index.shtml) and are available to the scientific community.

**Polymorphic analysis**. Polymorphisms of X-chromosome linked genes between JF1 (J) and C57BL/6N (B) were detected by RFLP analysis. Primers and restriction enzymes used for this are listed in Supplementary Table 6. $Xist^{+/-}$ MEFs were maintained in Dulbecco's modified Eagle's medium containing 10% foetal bovine serum (GIBCO, USA) at 37 °C in humidified 5% $CO_2$ in air. Total RNA was isolated using TRIzol reagent (Invitrogen, USA). RFLP experiments on (J × B) or (J × KO) F1 embryos (E9.5) were performed as described[16,23]. To analyse allelic expression in E13.5 $Ftx$ mutant eyes, 8–12-week-old $X^{GFP}$ female mice[24] were mated with WT (J × B) F1 hybrid male mice or $Ftx$-deficient male mice back-crossed to the JF1 genetic background. In our analysis of E13.5 embryos, single cells were recovered by trypsin treatment of the E13.5 eyes from which the pigmented epithelial layer had been removed surgically. The eGFP-positive cells were sorted by FACS on a MoFlo XDP cell sorter (Beckman Coulter, USA). Fragments of $Mecp2$ and $Ogt$ digested with the restriction enzymes listed in Supplementary Table 6 were analysed on an Agilent 2100 Bioanalyzer (Agilent Technologies, USA) in conjunction with the DNA 1000 assay, using the standard settings for data analysis in the instrument software.

**RT–qPCR analysis**. Total RNA was extracted using RNeasy micro kits (QIAGEN, Germany), and contaminating DNA was digested using RNase-free DNase (Nippon Gene, Japan). For reverse transcription, cDNA was synthesized using Super-Script III (Invitrogen, USA) with random hexamers. Primers and TaqMan assays used for RT–qPCR analysis of miRNA and several genes are listed in Supplementary Table 7. For miRNA expression analysis, total RNA was reverse-transcribed using TaqMan MicroRNA Reverse Transcription kits (Applied Biosystems, ABI, USA). Mature miRNAs were amplified with TaqMan Universal PCR Master Mix II (ABI, USA), and other genes were amplified with TaqMan Gene Expression Master Mix (ABI, USA) using a LightCycler 480 Instrument (Roche, Switzerland). All RT–qPCR reactions were performed using samples collected from three independent preparations.

**Histology**. Embryos removed from pregnant mice were fixed in Superfix (Kurabo, Japan) and processed for preparing paraffin wax sections. Embedded embryos were sectioned at 5 µm, stained with HE, and observed using bright-field microscopy. Eye size was measured at the largest diameter using Image J image analysis software (NIH, USA; https://imagej.nih.gov/ij/).

**Microarray analysis**. Eleven $Ftx^{-/-}$, three $Ftx^{-/}$Y, four WT female and three WT male eyes were analysed. Forty-two duplicate hybridization experiments were carried out. In E13.5 mouse embryos, eyes from which the pigmented epithelial layer had been removed surgically were collected for analysis. Total RNA was extracted from each eye sample using RNeasy micro kits (QIAGEN, Germany). Eluted RNA was quantified using an Agilent 2100 Bioanalyzer instrument, and 200 ng was used for amplification and labelling using Low Input Quick Amp Labeling kits (#5190-0442, Agilent Technologies, USA) following the manufacturer's instructions. Cy3-labelled cRNA was purified using RNeasy mini spin columns. The amount of labelled cRNA and uptake of cyanine were measured using a Nanodrop ND-1000 spectrophotometer (Thermo Fisher Scientific, USA). The labelled probes were hybridized to an Agilent 8 × 60 K whole-mouse genome microarray (G4852A). After washing, microarrays were scanned using an Agilent Microarray scanner. The data were normalized using the R statistical package with the "qspline" function of the "affy" package in Bioconductor software (https://www.bioconductor.org/install/).

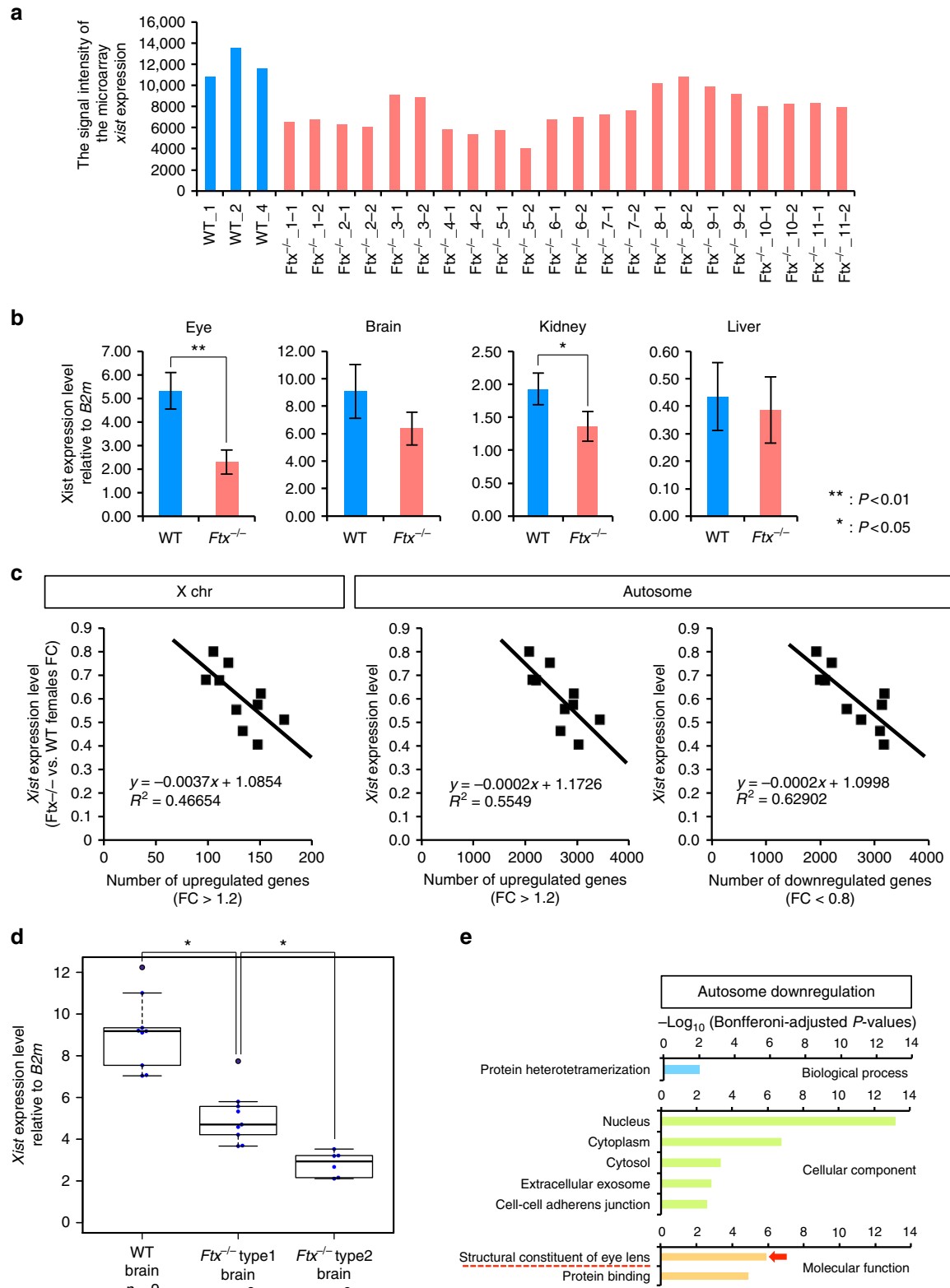

**Probe synthesis for RNA–FISH**. Probes were labelled fluorescently using nick translation kits (Abbott Molecular, USA) with Cy3-dCTP (PerkinElmer, USA) or Green-dUTP (Enzo Life Sciences, USA). *Xist*-specific probes were prepared from p*Xist*1986-9498. For other X-linked genes, specific probes were prepared from bacterial artificial chromosome (BAC) clones (BAC clone IDs: *Tmem29*:RP23-284M10, *Mecp2*:RP23-396E2 and *Ogt*:RP23-440J21).

**RNA–FISH**. The protocol was as reported[25]. Briefly, the eyes, brain and kidneys from E13.5 mouse embryos were dissected and dissociated into single cells by

pipetting in 1 × Acctase for 1 min at 25 °C. Cells were spun onto slides using a Cytospin 4 centrifuge (Shandon, UK) at 800 rpm for 4 min. Slides were rinsed in phosphate-buffered saline (PBS) for 15 s and then incubated in 0.1% Triton X-100 in PBS for 30 s. Slides were then fixed with 4% paraformaldehyde in PBS for 10 min and dehydrated through 70% and 100% ethanol. They were dried and hybridized to probes overnight at 42 °C. After hybridization, slides were washed in 50% for-mamide plus 2 × saline sodium citrate (SSC) buffer at 42 °C for 2 × 5 min and 0.1% Tween 20 plus 2 × SSC at 42 °C for 2 × 5 min. Then, slides were dehydrated through 70% and 100% ethanol, mounted in Vectashield with DAPI (Vector Labs, USA), and observed using an LSM710 confocal laser scanning microscope (Carl

**Fig. 6** Partial failure of XCI correlated with diminished *Xist* expression in *Ftx⁻/⁻* mice. **a** Diminished expression of *Xist* was detected in *Ftx⁻/⁻* females ($n = 11$; duplicate experiments). In the microarray analysis, seven *Xist* probes were analysed. They showed similar expression patterns, and the average value of these probes showed significant reduction of *Xist* expression in *Ftx⁻/⁻* eyes ($P = 0.015$). Representative values from probe "A_30_P01022001" are shown. The data for other probes are listed in Supplementary Data 1. **b** RT–qPCR validation of diminished *Xist* expression in *Ftx⁻/⁻* females at E13.5. All examined tissue samples—eyes, brains, kidneys and livers—were obtained from three different embryos. Significant differences were detected between WT ($n = 3$) and *Ftx⁻/⁻* ($n = 3$) mice using a two-tailed, unequal variance $t$-test (*$P < 0.05$; **$P < 0.01$). Error bars, s.d. *B2m*, beta-2 microglobulin. **c** Scatter plot showing the correlation between the *Xist* expression level and the numbers of upregulated and downregulated genes in *Ftx⁻/⁻* (*Ftx⁻/⁻* vs. WT, FC > 1.2 or FC < 0.8). The figures show the correlation between *Xist* expression level and the numbers of upregulated X-linked genes (FC > 1.2, left panel), the numbers of upregulated autosomal genes (FC > 1.2, middle panel), and the numbers of downregulated autosomal genes (FC < 0.8, right panel), respectively. **d** *Xist* expression was the most severely diminished in the brains of the *Ftx⁻/⁻* type 2 mutants. Diminished expression of *Xist* was detected in the brains of *Ftx⁻/⁻* female embryos at E13.5. Significant differences were detected between WT ($n = 9$) and *Ftx⁻/⁻* type 1 mutants ($n = 9$), and also between *Ftx⁻/⁻* type 1 mutants ($n = 9$) and *Ftx⁻/⁻* type 2 mutants ($n = 6$) using a two-tailed, unequal variance $t$-test (*$P < 0.01$). The box represents the median, 25th and 75th percentiles; whiskers indicate the maximum and minimum values. **e** Gene ontology analysis using autosomal downregulated genes (Bonferroni-adjusted $P$-values)

Zeiss, Germany). The *Xist* cloud size was measured using Image J image analysis software (NIH, USA).

**Whole-mount 3D RNA–FISH combined with immunofluorescence**. cDNA probes for *Mecp2* RNA detection were prepared as follows. Single-stranded RNAs corresponding to Macp2 transcripts (for Mecp2 probe-1: 74062552–74055116 bp and for Mecp2 probe-2: 74084953–74079723; NC_000086.7) were synthesized by in vitro transcription using T7 RNA polymerases (Roche, Germany). Using the in vitro-transcribed RNAs as a template, fluorescence-labelled DNA probes incorporating Cy3-dCTP (GE Healthcare, USA; PA53021) were generated by random-primed reverse transcription (RT) using SuperScript III reverse transcriptase (Life Technologies Corporation, USA). After the RT reaction, the template RNAs that remained in the reaction were destroyed by adding 2 μL of 4 M NaOH followed by incubation at 37 °C for 30 min, and then neutralized by adding 2 μL of 4 M HCl. After ethanol precipitation using ammonium acetate, the probe DNA was dissolved in 20 μL of formamide and stored at 4 °C in the dark. Whole embryos were dissected, treated with 0.1% Triton X-100 in PBS for 2 min on ice for permeabilization, and fixed with 4% paraformaldehyde in PBS with 0.1% Triton X-100 for 10 min at room temperature. Then, the samples were incubated sequentially in 2 × saline sodium citrate (SSC) buffer with 0.05% Tween 20; 2 × SSC and 25% formamide with 0.05% Tween 20; and 2 × SSC and 50% formamide with 0.05% Tween 20. After prehybridization in hybridization buffer (2 × SSC, 2 mg/mL of BSA and 50% formamide), hybridization was performed by incubation in a pre-denatured (70 °C for 10 min) probe mix (2 × SSC, 2 mg/mL of bovine serum albumin, BSA, 0.1 μg/μL of mouse Cot-1 DNA, Xist probe, Mecp2 probes 1 and 2, and 50% formamide) at 37 °C overnight. After hybridization, the samples were washed sequentially with a solution containing 2 × SSC, 50% formamide and 0.05% Tween 20 at 37 °C; with 2 × SSC containing 0.05% Tween 20 at 37 °C; and then with PBS containing 0.05% Tween 20 (PBST) for 5 min at room temperature. The samples were refixed with 4% paraformaldehyde in PBST for 5 min, and then subjected to immunofluorescence experiments. After incubation in blocking buffer (3% BSA and 5% donkey serum in PBST), the samples were reacted with the primary antibody against Pou5f1 (Oct4) (1:300 goat polyclonal anti-Oct-3/4 (N-19; Santa Cruz Biotechnology Inc., USA; sc-8628)) in blocking buffer at room temperature for 90–180 min or at 4 °C overnight. After washing with PBST, the samples were incubated with the secondary antibody conjugated with Alexa Fluor 647 (1:500 Alexa Fluor 647 donkey anti-goat IgG (H + L)) (Life Technologies Corporation, USA; ab150131) in the blocking buffer at room temperature for 90–180 min or at 4 °C overnight. After rinsing with PBST and post-fixation in 4% paraformaldehyde in PBST, the nuclear DNA of the samples was stained with DAPI and the samples were imaged using a confocal fluorescence microscope (LSM780; Carl Zeiss, Germany). Optical sections of whole embryos were imaged at intervals of 0.8 μm and were used to reconstruct images.

**Statistics and general methods**. The sample size of each experiment is shown in the figures or in the corresponding figure legends. The sample size was chosen based on previous experience in the laboratory, so that each experiment would yield a power that was sufficient for the detection of specific effects. No statistical methods were used to predetermine sample size. For in vivo experiments using mouse models, animals were chosen based on genotype. In Fig. 3b, foetal resorptions, commonly observed in both WT and mutant animals, were excluded from the count. In Fig. 2g, no data were excluded from the counting analysis performed in adult mice. In RNA–FISH experiments, the *Ftx*-deficient eyes from different litters were allocated randomly to the experimental groups, and WT eyes from different litter-mates were allocated randomly to the control groups. Investigators were not blinded to mouse genotype during the experiments. The detailed statistical conditions that were used in each experiment are described in the figure legends or in Methods section. All experiments were performed on at least three biologically independent samples. The data are expressed as the mean ± s.d. The unequal variance $t$-test was used for the statistical analysis of microarray results,

RT–qPCR and measurements of eye size, to analyse the significance of any differences. All analyses were two-tailed. The $\chi^2$-test was used for the statistical analysis of mating experiments and of the incidence of each type of eye abnormality. $P < 0.05$ and/or $P < 0.01$ were considered statistically significant.

## Data availability

All microarray data that support the finding of this study have been deposited in the National Center for Biotechnology Information Gene Expression Omnibus (GEO) with the accession number (GSE100700). The *miRNA374/421* doubly deficient mice are available to the scientific community from RIKEN BioResource Center, upon request (BRC No. RBRC 09530). All other data that support the findings of this study are available from the corresponding author upon reasonable request.

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

## Acknowledgements

We thank Dr M. Sugimoto at RIKEN BioResource Center for providing the p*Xist* 1986-9498 plasmid, Y. Esaki and S. Nishioka for technical assistance in generating *miRNA374/421* doubly deficient mouse lines, and A. Hirano for the maintenance of animals. We also thank Drs K. Ohno and Y. Kuroda at Tokyo Medical and Dental University for discussion on human diseases, and Drs S. Nakagawa, M. Okabe and D. Endo for critical reading and discussion. This work was supported by the Japan Science and Technology Agency, Precursory Research for Embryonic Science and Technology (PRESTO) and by a Grant-in-Aid for Scientific Research from the Ministry of Education, Culture, Sports, Science, and Technology [23500492, 15H01468, 26430087, 17H05597, 17K07498, and 18H04892] to S.K. The funding bodies had no roles in the study design, data collection and analysis, decision to publish or preparation of the manuscript.

## Author contributions

Y.H., M.S. and S.K. designed the research. Y.H., M.S., H.S. and S.K. performed the research, assisted by H.H., T.S., K.A., T.K. and F.I. The KO mice were generated by S.K. and H.H., Y.H., M.S. and S.K. analysed the data. Y.H. and S.K. wrote the manuscript, and all authors discussed the results and commented on the manuscript.

## Additional information

**Competing interests:** The authors declare no competing interests.

