## [Peer Review File · Nature Communications]

Reviewers' comments:

Reviewer #1 (Remarks to the Author):

[Editorial Note: Please see reviewer comments on the following page]

In this manuscript, Hosoi *et al* set out to assess the role of the long non-coding RNA Ftx during random X-chromosome inactivation (rXCI) in mouse. Using a combination of mouse genetics, microarray analysis and RNA-FISH, the authors propose that Ftx could be involved in the rXCI mechanism *in vivo* and that its absence partially impairs Xist expression, dosage compensation and subsequently eye development in a small portion of Ftx^{-/-} females.

The findings are of interest for our better understanding of rXCI and the consequences of its partial defect. However, several missing experiments dampen my enthusiasm for the paper and I believe that the conclusions are greatly over-stated, in particular the Figure 6f model, for which the authors provide no evidence. Overall the authors must tone down the text in the Abstract and throughout the m/s.

Major comments

Ftx role in rXCI

The most important message of the paper is the potential involvement of Ftx as a player of rXCI through Xist upregulation.

In absence of Ftx, the authors showed decreased expression of Xist in eye, kidney and a tendency in the brain of E13.5 female embryos (Figure 6c).

Xist expression seems very variable in the different mutant eyes (Figure 6a). Could that be related to the “Type” of the mutant embryos?

The authors should discuss why they see 40% decrease of Xist by RT-qPCR and no difference in the Xist clouds by RNA-FISH (Figure 5). The authors should try to quantify the Xist cloud area in wt and Ftx^{-/-} samples.

X-linked gene analysis by RNA-FISH has revealed a perturbation of the monoallelic status of 3 candidate genes in the eye, brain and kidney (Figure 5). Between 11% and 26% of the cells are showing either biallelic expression or monoallelic expression from the inactive X. The authors failed to discuss how the genes could become only expressed from the inactive X chromosome? How could that fit with their model?

How could this very small subset of biallelic cells explain the upregulation of the X-linked genes observed by microarray? The authors should provide evidence that the overexpression is coming from the inactive X and not from the active X chromosome to support their hypothesis of Xi silencing impairment.

Eye phenotype and impairment of dosage compensation:

The authors made the interesting observation that a small percentage (20-25%) of Ftx^{-/-} female embryos developed Type 2 and 3 eye abnormalities.

They then tried to link this phenotype to dosage compensation impairment. Their claims are only supposition, as they only provided gene expression and RNA-FISH analysis of Ftx^{-/-} “normal eye” (Type 1) mice and not the ones with eye phenotype (Type 2 and Type3).

If the authors want to show that the eye phenotype is linked with defect in X-linked gene silencing, they should at least provide RNA-FISH for *Tmem29*, *Mecp2* and *Ogt* in Type 2 eye sections and show a large increase in biallelic expression for those genes. It would be very appreciated that the authors include a control of a non upregulated gene in their RNA-FISH.

A control of RNA-FISH efficiency should also be included such as male cells and/or a constitutive escapee gene. The authors are dissociating the tissues into single cells and then sticking them to a coverslip rather than directly doing RNA-FISH on eye section. It is then very important to control the quality of the experiment.

Initiation or maintenance?

The authors provide a model in which they suggest that, in absence of *Ftx*, *Xist* is downregulated and could not target all the X-linked genes on the X chromosome and then fail to induce their silencing.

This claim is strongly lacking evidence. The authors must remove this model from the m/s.

No experiment is provided to understand if *Ftx* is important for the initiation or the maintenance of rXCI. This is a key question to understand the mechanism of rXCI. Are the X-linked genes upregulated at E13.5 ever silenced?

The authors should provide RNA-FISH on sections of *Ftx*^{-/-} embryos when rXCI is initiated (or soon after, *eg* E6.5) and compare them with their data at E13.5.

Ftx could also be involved in repressing genes that more easily tend to escape in a tissue or temporal specific manner. Could the authors compare their list of upregulated X-linked genes to such conditional escapee list?

And do the upregulated X-linked genes share some specific features? Lowly expressed genes? Located in first regions bound by *Xist* as described by the Guttman lab? Close to repeats?

Minor comments

1. The authors claimed that Type 1 *Ftx*^{-/-} embryos do not present eye phenotype, however several autosomal genes involved in eye development are down-regulated (GO analysis). In Figure 3d, *Ftx*^{-/-} Type 1 eyes are statistically smaller than wt at E15.5. Did the authors check that these mice do not develop visual defects later on?
2. What is the statistical test used for the FC expression in the microarray? And which threshold has been used to extract the up and down regulated genes? P-value<0.05?
A list of all the X-linked genes, their FC and p-value should also be provided in a Supplemental Table.

3. In Figure 4c, the authors failed to validate 2 genes (in a total of 5) and argue that it is presumably due to their low expression level. *Slc16a2* does not seem to be lowly expressed compared to *Tmem29* and *Mecp2*.
4. The Figures 5b and 5c are misleading and give the impression that 100% of the observed cells showed a FISH signal for the analysed genes. Is that true? Percentage of cells without a signal for the X-linked gene should be provided and included in the graph.
5. Kidney and Brain tissues show similar down-regulation of *Xist* and misregulation of X-linked genes by RNA-FISH at E13.5. The authors should provide evidence of absence of phenotype in these tissues or remove their conclusion about the eye "being the most sensitive tissue" (line 225).
Based on their RNA-FISH, they should not claim that misexpression is "the same or lesser" (line 225) in brain and kidney compared to eye, as it seems just similar. Or they should use an appropriate statistical test.
6. Several X-linked genes are expressed in the brain, such as *Mecp2*, which is linked with Rett syndrome in Humans. Did the *Ftx*^{-/-} females exhibit any neuronal development defect after birth?
7. The authors should provide the unit for the y axis in Figure 6a.
8. What is the difference between Figure 6b and 6c (eye)? And why are the graphs not similar?
9. Care should be taken in the manuscript to provide the exact number of genes, percentages... rather than "some mice", "a subset of".
10. Typo on Figure 4a: microarray.

Reviewer #2 (Remarks to the Author):

Hosoi et al. in their manuscript NCOMMS-17-24595A-Z entitled "Female mice lacking Ftx lncRNA exhibit impaired chromosome inactivation and a microphthalmia-like phenotype" investigate the function for Ftx lncRNA in somatic tissues and random XCI. Towards this goal, the authors examine the adult Ftx KO female and male mice and detect an eye development defect resembling human microphthalmia. Analyses of the mouse cohort for 2-3 generations suggest no significant inheritance pattern of this pathology. The authors conclude that it is a stochastic process that impacts only a subset of female Ftx^{+/-} and Ftx^{-/-} mice. The authors previously determined that targeting of Ftx locus (Exons 1-4) also leads to altered expression of 2 miRNAs that are encoded within Ftx intron 12. To address if the Ftx mutant mice eye phenotype is due to loss of miRNAs 374 and 421, they generate miRNA374/421 KO knockout mice and determine that none of the miRNA374/421 KO mice present pathological abnormalities. By performing microarray analyses using E13.5 eye samples from WT and Ftx mutant female and male mice, they further argue that Ftx loss induces differential change in both X-linked and autosomal gene expression in both female and male Ftx mutant mice. RNA FISH analyses suggest that 3 of the upregulated X-linked genes present biallelic expression pattern in female Ftx^{-/-} eye samples compared to control samples. Even though the authors detect intact Xist RNA clouds in cells isolated from female Ftx^{-/-} eye samples, RT-PCR analyses shows ~30% reduction in Xist expression. Based on these data, a model was proposed in which Ftx loss results in downregulation of Xist expression, which in turn trigger upregulation of X-linked genes that subsequently lead to the microphthalmia-like phenotype. I believe determining the functional relevance of lncRNAs by using mouse models is important. However, I unfortunately do not agree with the authors' conclusions and that the findings of this paper provide strong evidence towards Ftx as a regulator of Xist or XCI. I base my decision on the following points:

1. The Ftx deletion in female mice only impact a subset of mice and do not present any significant inheritance pattern. Strong XCI defects couple with penetrant phenotypes which should get transmitted in the progeny. Examining the cohort beyond F3 might be helpful, but I believe it is more important to investigate if Ftx KO founder mice have any genome-wide genetic alterations that might have contributed to the pathology.
2. It is not clear which Cre mouse was used to generate Ftx or miRNA374/421 DKO mice.
3. Fig 1SD. Ftx expression in the eye samples should be examined in addition to the kidney samples that are reported in this figure. More importantly, Ftx expression in WT female mice samples needs to be reported so one can evaluate the affect of Ftx heterozygous and homozygous deletion on Ftx expression.
4. Fig 2d-e. Southern blotting for female homozygous mutant miRNA374/421 mice is missing. More importantly in panel e, female WT levels for miRNAs 374 and 421 were not reported. Interestingly, in an earlier manuscript the authors show that male blastocysts lack miRNA374 and 421 expression, which is present in kidney samples from male mice. This data argue that Ftx and miRNA374/421 expression might be differently regulated in somatic cells. It is therefore essential to evaluate Ftx levels in eye samples from miRNA374/421 DKO mice.
5. Fig 2f. I find inconsistencies in the animal numbers presented in Figure 2f and those reported in Table S1 and S2, which will likely impact significance values presented in Fig 2f. Differences are as follows:
Table S1. Male Ftx ^{-Y}, n=103 (reported in Fig 2f) vs n=129 (Table S1). Female Ftx ^{+/-}, n=118 (reported in Fig 2f) vs n=127 (Table S1).
Table S2. Female miRNA control, n=35 (reported in Fig2f) vs n=59 (Table S2). Female miRNA ^{+/-} KO,

n=69 (reported in Fig 2f) vs n=45 (Table S2).

6. Fig 3b. Animal numbers used to plot Fig 3b should be reported. More importantly, WT and mutant animal numbers vary significantly. Especially in a study like this where a stochastic phenotype is detected, I think it is important to compare littermates and I wonder if data reported in this figure were collected considering this fact.

7. Fig 3d. How does the statistics in Fig 3d correlate with Fig 3c data? There is inconsistency in the numbers especially for the % of female mice that present type 2 and 3 pathology.

8. Fig 4a. Were the mice used in microarray analyses, littermates? Microarray data was only supplemented as gene enrichment tables instead of individual gene expression changes. 4b. X-linked and autosomal gene expression changes should be presented as a scatter plot comparing wt and mutant samples. It is very difficult to interpret microarray results based on the short list provided for selected X linked genes in Fig 4d. The threshold applied on microarray data are at the border and might suggest no change. Detecting 1.2-1.7 fold change might purely be a reflection of sample size. Applying more stringent cut offs would be necessary. If 2-fold change is applied, only 3 genes on X show upregulation (considering this list is complete), which to this reviewer does not signify substantial XCI defect. These observations argue for a non-XCI related process in explaining the Ftx mutant mice pathology.

9. Fig 4b. How do the authors explain X linked gene upregulation in male Ftx mutant mice?

10. Fig 4e. Should be replaced with allele specific qRT-PCR analyses. Alternatively, Sanger Sequencing of gDNA and transcript analysis of cDNA at these loci should be done to validate allele specific transcription changes due to Ftx deletion.

11. Fig 5. Total cell numbers analysed, and the cell numbers that present mono- or biallelic expression for 3 X linked genes were not reported. The authors indicate that they can detect Xist RNA cloud in 85% of cells. I wonder if this was factored in in their biallelic RNA-FISH analyses in panel b or c.

12. Fig 5a-b-c. I am confused with the results reported here. For example, cells that are categorized as light pink are supposed to show expression of X-linked genes that only express from Xi. If this data is correct, does it mean Ftx cause loss of active X chr specific expression of these genes as well? This reviewer thinks this pattern might be reflective of those cells in which RNA-FISH did not work efficiently. It is also important to show that the second RNA signal detected for X-linked genes is indeed coming from the second X chr copy. Were wt and mutant eye samples collected from littermates? Have the authors examined karyotype of these mice? Were there any chromosomal abnormalities?

13. Xist expression looks intact in FISH analyses (Fig 5a). However, the authors show by RT-PCR that there is ~30% decrease in Xist transcript. This reviewer thinks that if the authors would like to suggest that this is causal to upregulation of X-linked genes, it is necessary to determine changes in heterochromatin marks associated with these loci and perform allele-specific RNA-Seq.

14. Based on these comments, this reviewer thinks that the XCI phenotype in female Ftx mutant mice is very mild to none and the data presented is not sufficient to support the working model of the authors.

Reviewer #3 (Remarks to the Author):

This manuscript by Hosoi et al., studies the effects of loss of Ftx in vivo. Ftx heterozygous and homozygous knockout female mice display an eye phenotype that resembles microphthalmia. This phenotype is not fully penetrant and only present in female offspring suggesting a relation to X chromosome inactivation. The authors indeed find reduced expression of Xist and concomitant up-regulation of X linked genes in mice with an eye phenotype. In addition, genes involved in eye development appear downregulated in affected females, suggesting a link between loss of Xist expression and reduced expression of genes involved in eye development. The findings are in line with previous reports on conditional Xist knockout mice showing various phenotypes. This study indicates that Ftx mutations appear to affect eye development more than other tissues and organs, suggesting an more important role for Xist expression in this tissue. I have several suggestions to improve the manuscript.

One major question that is not addressed by the authors is what is really going wrong. Is X chromosome inactivation initially random and initiated normally followed by loss of X chromosome inactivation, or is initiation of X chromosome inactivation affected? To address this question studies on early embryo's need to be performed to test whether loss of XCI is present from the start with indicates that initiation is affected.

In this respect it will be important to test whether XCI is random, for instance by generating F1 B6: mollosinus mice. I think this is important to know, because the phenotype might be hard to explain if X chromosome inactivation will be skewed towards inactivation of the wild type X chromosome. Second the Ftx knockout allele was generated in a 129Sv background and was then backcrossed into the B6 background. As the Ftx mutation is linked to the X inactivation center, and the 129/Sv and B6 strains retain different Xce strengths it is expected that Ftx+/- heterozygous mice might display skewed XCI which may affect the phenotype.

In the final paragraph the effects of Ftx homozygous mutations on Xist and X linked and autosomal gene expression are studied, showing reduced Xist expression and concomitant effects on both X linked and autosomal genes. Here I would have liked to see the same effects in the affected heterozygous mice.

In the discussion it is concluded that 'So far, there have been no reports of mutantother than embryonic death'. This is not completely true as the Lee lab also deleted Xist specifically in the haematopoietic lineage resulting in loss of XCI and anaemia.

Is there an overlap in the limited subset of X linked genes that are affected in Ftx-/- homozygous knockout mice?

Figure 2e, why was Ftx expression not examined?

Where are Ftx and Xist in Figure 4C, these genes need to be indicated.

Figure 6a, I suppose this is data obtained on RNA isolated from the eye.

We would like to thank the reviewers for their constructive comments, which have helped to improve the content of our paper. We have revised our manuscript accordingly. The new text is highlighted in yellow in the revised manuscript.

While our manuscript has been under revision, independently from our study, Furlan, G et al. reported that *Ftx* is required both in the upregulation of *Xist* and in control X-inactivation using ES cells (Furlan G *et al.*, Mol Cell 70, 1-11, 2018). We think their *in vitro* study and our *in vivo* study complement each other and the importance of *Ftx* in random XCI has been clarified in depth by the combination of these two papers.

We have added the above sentence at the bottom of our revised manuscript.

Reviewer #1 (Remarks to the Author):

please see attached document

In this manuscript, Hosoi et al set out to assess the role of the long non-coding RNA Ftx during random X-chromosome inactivation (rXCI) in mouse. Using a combination of mouse genetics, microarray analysis and RNA-FISH, the authors propose that Ftx could be involved in the rXCI mechanism in vivo and that its absence partially impairs Xist expression, dosage compensation and subsequently eye development in a small portion of Ftx^{-/-} females. The findings are of interest for our better understanding of rXCI and the consequences of its partial defect. However, several missing experiments dampen my enthusiasm for the paper and I believe that the conclusions are greatly over-stated, in particular the Figure 6f model, for which the authors provide no evidence. Overall the authors must tone down the text in the Abstract and throughout the m/s.

Answer: We appreciate the reviewer's positive view. To make the impairment status of random XCI in KO mice clearer, we have performed additional experiments, which yielded the following results.

- 1) RFLP analysis at E13.5 revealed ectopic expression of X-linked genes from the inactive X, suggesting that *Ftx* is involved in random XCI and acts in cis.
- 2) RNA-FISH at E6.5 showed that random XCI was impaired in its early stages in *Ftx* KO embryos, suggesting its potential role in the initiation of random XCI.
- 3) Allelic expression analysis of X-linked genes at E9.5 showed that deleting *Ftx* on one X chromosome did not skew XCI toward the WT X chromosome.
- 4) Quantification of *Xist* expression in the type 2 brain supported the idea that there could be a relationship between reduced *Xist* expression and the severity of the variable phenotype of these KO mice.

Following the referee's comment, we have eliminated the model presented in Fig. 6f, and toned down the

text in the abstract and throughout the manuscript. We would like to submit our analysis of the precise mechanism as a separate report in the future.

Major comments

Ftx role in rXCI

1, The most important message of the paper is the potential involvement of Ftx as a player of rXCI through Xist upregulation. In absence of Ftx, the authors showed decreased expression of Xist in eye, kidney and a tendency in the brain of E13.5 female embryos (Figure 6c). Xist expression seems very variable in the different mutant eyes (Figure 6a). Could that be related to the “Type” of the mutant embryos?

Answer: We think it highly probable that *Xist* expression level is related to the “Type” of *Ftx*^{-/-} mutant embryos, because our recent findings are that the *Xist* expression level in the type 2 brain was more severely diminished than those in type 1 and WT embryos. In this experiment, there was no appropriate method for measuring the expression of *Xist* in affected eyes. Therefore, instead of using the eyes, we measured *Xist* expression in the brain tissues of type 2 embryos and compared it with those of type 1 and WT embryos. This analysis is based on our observation that the sections of type 2 brains appeared indistinguishable from type 1 and WT brains, as far as we could tell. From this, we found that *Xist* expression levels in brain tissue were more severely diminished in type 2 mutants than in type 1 mutants and WT embryos (new Fig. 6d), suggesting that the phenotypic severity of eye defects was correlated with the variability of diminished *Xist* expression in tissues other than the eye. This result has been added to the revised manuscript: lines 289-295.

2, The authors should discuss why they see 40% decrease of Xist by RT-qPCR and no difference in the Xist clouds by RNA-FISH (Figure 5). The authors should try to quantify the Xist cloud area in wt and Ftx^{-/-} samples.

Answer: Following the reviewer’s comment, we quantified the *Xist* cloud area in WT and *Ftx*^{-/-} samples. As shown in the revision (lines 219-222, and the new Fig. S4), the *Xist* clouds were slightly—but significantly—smaller than those of the WT cells in all examined tissues.

3, X-linked gene analysis by RNA-FISH has revealed a perturbation of the monoallelic status of 3 candidate genes in the eye, brain and kidney (Figure 5). Between 11% and 26% of the cells are showing either biallelic expression or monoallelic expression from the inactive X. The authors failed to discuss how the genes could become only expressed from the inactive X chromosome? How could that fit with their model?

How could this very small subset of biallelic cells explain the upregulation of the Xlinked genes observed by microarray? The authors should provide evidence that the overexpression is coming from the inactive X and not from the active X chromosome to support their hypothesis of Xi silencing impairment.

Answer: We appreciate this comment. We have removed the model in our original Fig. 6f. In this

manuscript, we have focused on our hypothesis that *Ftx* could have roles in random XCI *in vivo* and that its deletion could cause eye abnormalities in female mice. Because Reviewer 2 had the same concern that overexpression could come from the inactive X chromosome, we carried out allelic expression analysis in our KO model. We found that X-linked genes were found to be misexpressed from inactive X chromosomes, suggesting that *Ftx* could have a role in XCI and could act in cis. However, we were unable to rule out the possibility that overexpression could also come from the active X in homozygous mutants. This result has been added to the revised manuscript, lines 223-240, and Figs S5–S7. We would like to study the mechanisms in more detail and report on them in another paper.

Eye phenotype and impairment of dosage compensation:

4, The authors made the interesting observation that a small percentage (20-25%) of $Ftx^{-/-}$ female embryos developed Type 2 and 3 eye abnormalities. They then tried to link this phenotype to dosage compensation impairment. Their claims are only supposition, as they only provided gene expression and RNA-FISH analysis of $Ftx^{-/-}$ “normal eye” (Type 1) mice and not the ones with eye phenotype (Type 2 and Type3). If the authors want to show that the eye phenotype is linked with defect in X-linked gene silencing, they should at least provide RNA-FISH for *Tmem29*, *Mecp2* and *Ogt* in Type 2 eye sections and show a large increase in biallelic expression for those genes. It would be very appreciated that the authors include a control of a non upregulated gene in their RNA-FISH.

Answer: We appreciate this comment. We took into consideration one possibility that the reviewer suggested, and we carried out RNA–FISH analysis of type 2 eyes. As shown below, the percentage of cells showing biallelic expression did not increase. Thus, we could exclude the possibility that the percentage of cells showing ectopic expression from Xi increased in affected type 2 eyes. This result (Fig. S3c) and comments have been added in the revised MS: lines 277-281. However, we offer a different explanation for the link between the phenotype and defect in X-linked gene silencing. As described in our original manuscript, lines 302–304 (also lines 354-355 in the revised manuscript), we propose that the more *Xist* was diminished, the greater the number of X-linked genes that were overexpressed, so that reduced *Xist* expression might result in a more severe phenotype. This is based on our observation of a negative correlation between the level of reduction in *Xist* expression and the number of upregulated X-linked genes in $Ftx^{-/-}$ female mice (Fig. 6c, left panel). Further, to support this hypothesis, we have added a new RT–qPCR result showing more diminished *Xist* expression in type 2 than in WT and type 1 brains (new Fig. 6f). All these data support our hypothesis. These results and further discussion have been added to the revised manuscript, lines 289-295, 353-354 and 357-359. Following the reviewer’s comment, we have also included a control, the non-upregulated *Pgk1* gene, in lines 216-219, new Fig. S3b. We have removed the model presented in the original Fig. 6f.

5, A control of RNA-FISH efficiency should also be included such as male cells and/or a constitutive escapee gene. The authors are dissociating the tissues into single cells and

then sticking them to a coverslip rather than directly doing RNA-FISH on eye section. It is then very important to control the quality of the experiment.

Answer: Following the reviewer's comment, we have included male cells as a control (lines 214-215, Fig. S3a). Because Reviewer 2 had the same concern about our RNA-FISH methodology, we agree that the original graph in Fig. 5 and lack of detailed explanation could have led to a misunderstanding regarding the results. We have replaced them with a new Fig. 5, added an explanation in the legend, and cited the original method reported by Tilghman's laboratory: lines 667-671 and 447.

Initiation or maintenance?

6, The authors provide a model in which they suggest that, in absence of Ftx, Xist is downregulated and could not target all the X-linked genes on the X chromosome and then fail to induce their silencing. This claim is strongly lacking evidence. The authors must remove this model from the m/s.

Answer: We appreciate this comment. We have eliminated the model presented in the original Fig. 6f in our revised manuscript.

7, No experiment is provided to understand if Ftx is important for the initiation or the maintenance of rXCI. This is a key question to understand the mechanism of rXCI. Are the X-linked genes upregulated at E13.5 ever silenced? The authors should provide RNA-FISH on sections of Ftx^{-/-} embryos when rXCI is initiated (or soon after, eg E6.5) and compare them with their data at E13.5.

Answer: Following the reviewer's comment, RNA-FISH analysis was carried out at E6.5. As a result, misexpression of X-linked genes was detected when random XCI was initiated, suggesting the involvement of *Ftx* in the initiation step of this process. Comparing the percentages of cells showing misexpression at E6.5 (2.7%) with that of E13.5 (9.7%), it is possible that *Ftx* also functions in the maintenance of random XCI. The results are summarized in our new Fig. S8 and comments have been added in lines 249-255 in the revised manuscript.

Ftx could also be involved in repressing genes that more easily tend to escape in a tissue or temporal specific manner. Could the authors compare their list of upregulated Xlinked genes to such conditional escapee list? And do the upregulated X-linked genes share some specific features? Lowly expressed genes? Located in first regions bound by Xist as described by the Guttman lab? Close to repeats?

Answer: We are also interested in whether upregulated genes might have some characteristics, such as conditional escapees in a tissue- or time-specific manner. As far as we examined and as described below, we could not find significant common characteristics among the upregulated genes. Following the reviewer's comments, we examined the points as follows. Among the 93 upregulated genes, five of them overlapped with the conditional escapees reported by Berletch et al. (PLoS Genet. 2015 Mar

18;11(3):e1005079. doi: 10.1371/journal.pgen.1005079). There seems to be no specific functional features among the upregulated genes (Table S3). These were not necessarily lowly expressed genes (see the new Fig. 1 below) but tended to be located in the first region bounded by *Xist* (see the new Fig. 2 below). This is known to be a gene-rich region, so we wonder if it has significance. We have attached the results for this response only for the reviewers: it is not included in the revised manuscript. As far as we examined from the escapee list available, not many genes overlapped between conditional escapees and upregulated ones. However, only a limited number of tissues (brain, spleen, ovary, and placenta) were analysed for conditional escapees, so it would be of interest to check these linkages in future.

Fig. 1 Microarray signal intensity of upregulated X-linked genes red: FC>1.2, blue: FC<0.8, black: 0.8<FC<1.2

Fig. 2 Mapping of *Xist* localization on the inactive X chromosome and X-linked upregulated genes in The *Ftx* KO

Minor comments

1. The authors claimed that Type 1 *Ftx*^{-/-} embryos do not present eye phenotype, however several autosomal genes involved in eye development are downregulated (GO analysis). In Figure 3d, *Ftx*^{-/-} Type 1 eyes are statistically smaller than wt at E15.5. Did the authors check that these mice do not develop visual defects later on?

Answer: We appreciate the reviewer's comment. As the reviewer pointed out, there might be visual defects in adult KO mice. Unfortunately, we have not yet examined this possibility, because specialized equipment is necessary for a detailed analysis. We would like to examine this in the near future.

2. What is the statistical test used for the FC expression in the microarray? And which threshold has been used to extract the up and down regulated genes? P value < 0.05? A list of all the X-linked genes, their FC and p-value should also be provided in a Supplemental Table.

Answer: We thank the reviewer for pointing this out. We used a two-tailed, unequal variance t-test for the analysis (WT *n* = 3 vs. KO *n* = 11; *P* < 0.05), and thresholds of FC > 1.2 for upregulated genes, and FC < 0.8 for downregulated genes. We have added this information in the legend of Figure 4, lines 648-649. We have provided a list of all the X-linked genes as a new Supplementary Table S8, lines 657-658.

3. In Figure 4c, the authors failed to validate 2 genes (in a total of 5) and argue that it is presumably due to their low expression level. *Slc16a2* does not seem to be lowly expressed compared to *Tmem29* and *Mecp2*.

Answer: We thank the reviewer for pointing this out. We thought the RT-qPCR result for *Slc16a2* was more reliable than the microarray result and that the DNA microarray might quantify the expression level as being lower than the actual level, probably because of the inappropriate probe design for this gene. We have rewritten the sentence to try to make this point clear: line 206.

4. The Figures 5b and 5c are misleading and give the impression that 100% of the observed cells showed a FISH signal for the analysed genes. Is that true? Percentage of cells without a signal for the X-linked gene should be provided and included in the graph.

Answer: Following the reviewer's comment, we have provided the percentages of cells without signals for *Xist* and/or X-linked genes, and have redrawn the graphs as new Figs 5b and 5c.

5. Kidney and Brain tissues show similar down-regulation of *Xist* and misregulation of X-linked genes by RNA-FISH at E13.5. The authors should provide evidence of absence of phenotype in these tissues or remove their conclusion about the eye "being the most sensitive tissue" (line 225). Based on their RNA-FISH, they should not claim that misexpression is "the same or lesser" (line 225) in brain and

kidney compared to eye, as it seems just similar. Or they should use an appropriate statistical test.

Answer: We agree that we could not exclude the possibility of phenotypes appearing in KO kidneys and brains. Following the reviewer's comment, we have removed line 225, and changed the wording of lines 244-246 in the revision.

6. Several X-linked genes are expressed in the brain, such as Mecp2, which is linked with Rett syndrome in Humans. Did the $Ftx^{-/-}$ females exhibit any neuronal development defect after birth?

Answer: We appreciate the reviewer's comment. As far as we have observed, there was no typical phenotype of Rett syndrome, such as arrested development and/or abnormal behaviour. However, there remains the possibility of deterioration of higher brain functions. We would like to examine the behaviour of these KO mice in the future.

7. The authors should provide the unit for the y axis in Figure 6a.

Answer: Following the reviewer's comment, we have added "the signal intensity of the microarray" to the Y-axis in Figure 6a.

8. What is the difference between Figure 6b and 6c (eye)? And why are the graphs not similar?

Answer: We think the difference may be due to the variability of the effects on KO eyes because the new Fig. 6f and Fig. 6a suggest that every KO mouse showed different effects on *Xist* reduction level. Although both Figs 6b and 6c in the old version showed significantly reduced expression of *Xist* and as there is no change to our argument, we have removed Fig. 6b to avoid confusion. We have also added information on the samples used to obtain the data in Fig. 6c in the legend, as follows: all examined tissue samples, eyes, brain, kidney, and liver, were each obtained from three different embryos.

9. Care should be taken in the manuscript to provide the exact number of genes, percentages... rather than "some mice", "a subset of".

Answer: Following the reviewer's comment, we have added the exact numbers in lines 122 and 272.

In the old line 122: "the abnormal eye phenotype appeared stochastically only in a subset of homozygous ($Ftx^{-/-}$) and heterozygous ($Ftx^{+/-}$, $Ftx^{-/+}$) female mice but not in hemizygous male mice."

In the new lines 120-124: "the abnormal eye phenotype appeared stochastically only in a subset of homozygous female ($Ftx^{-/-}$) with 5 (Fig. 1b) and 11 (Fig. 1c) mice affected. Three heterozygous female ($Ftx^{+/-}$, $Ftx^{-/+}$) mice were affected (Fig. 1b) but no hemizygous male mice were affected (Fig. 1b, c)."

Old line 272; "some female mice"

New lines 316-318: "Given that microphthalmia occurred only in some of the female mice (17-26 of homozygous and 7.1-10% of heterozygous mutants) and barely at all in male mice".

10. Typo on Figure 4a: microarray.

Answer: We thank the reviewer for pointing this out. We have corrected the typo.

Reviewer #2 (Remarks to the Author):

Hosoi et al. in their manuscript NCOMMS-17-24595A-Z entitled “Female mice lacking Ftx lncRNA exhibit impaired chromosome inactivation and a microphthalmia-like phenotype” investigate the function for Ftx lncRNA in somatic tissues and random XCI. Towards this goal, the authors examine the adult Ftx KO female and male mice and detect an eye development defect resembling human microphthalmia. Analyses of the mouse cohort for 2-3 generations suggest no significant inheritance pattern of this pathology. The authors conclude that it is a stochastic process that impacts only a subset of female Ftx^{+/-} and Ftx^{-/-} mice. The authors previously determined that targeting of Ftx locus (Exons 1-4) also leads to altered expression of 2 miRNAs that are encoded within Ftx intron 12. To address if the Ftx mutant mice eye phenotype is due to loss of miRNAs 374 and 421, they generate miRNA374/421 KO knockout mice and determine that none of the miRNA374/421 KO mice present pathological abnormalities. By performing microarray analyses using E13.5 eye samples from WT and Ftx mutant female and male mice, they further argue that Ftx loss induces differential change in both X-linked and autosomal gene expression in both female and male Ftx mutant mice. RNA FISH analyses suggest that 3 of the upregulated X-linked genes present biallelic expression pattern in female Ftx^{-/-} eye samples compared to control samples. Even though the authors detect intact Xist RNA clouds in cells isolated from female Ftx^{-/-} eye samples, RT-PCR analyses shows ~30% reduction in Xist expression. Based on these data, a model was proposed in which Ftx loss results in downregulation of Xist expression, which in turn trigger upregulation of X-linked genes that subsequently lead to the microphthalmia-like phenotype. I believe determining the functional relevance of lncRNAs by using mouse models is important. However, I unfortunately do not agree with the authors’ conclusions and that the findings of this paper provide strong evidence towards Ftx as a regulator of Xist or XCI. I base my decision on the following points:

Answer: We thank the reviewer for the positive view of the functional importance of Ftx lncRNA *in vivo*.

To clarify the impairment status of random XCI in Ftx KO mice, we have performed additional experiments that provided the following outcomes.

- 1) RFLP analysis at E13.5 revealed ectopic expression of X-linked genes from the inactive X, suggesting Ftx was involved in random XCI and acts in cis.
- 2) RNA-FISH at E6.5 showed that random XCI was impaired in its early stages in Ftx KO embryos, suggesting its potential role in the initiation of random XCI.
- 3) Allelic expression analysis of X-linked genes at E9.5 showed that deleting Ftx on one X chromosome did not skew XCI toward the WT X chromosome.

- 4) Quantification of *Xist* expression in type 2 brains supported the idea that there could be a relationship between reduced *Xist* expression levels and the severity of the variable phenotype of these KO mice.

These results are summarized in new Figs 6f and S4–S10 and are mentioned in the Results and Discussion (lines 223-240, 249-255, 255-262 and 289-296). We hope the revised manuscript will satisfy the reviewer's concerns.

1. The *Ftx* deletion in female mice only impact a subset of mice and do not present any significant inheritance pattern. Strong XCI defects couple with penetrant phenotypes which should get transmitted in the progeny. Examining the cohort beyond F3 might be helpful, but I believe it is more important to investigate if *Ftx* KO founder mice have any genome-wide genetic alterations that might have contributed to the pathology.

Answer: We thank the reviewer for this suggestion. We agree that the stochastic phenotype of *Ftx* KO mice showed an unusual inheritance pattern which seems difficult to explain by conventional genetics. Based on these observations, we suppose that the reviewer is requesting that we carry out whole-genome sequencing using next-generation sequencing to show any functional linkage between the phenotype and *Ftx* disruption. We could understand the importance of searching for genome-wide genetic alterations if the KO mice had been generated using the CRISPR/Cas9 system. In such cases, the chances of off-target gene modifications must be considered. However, we generated the *Ftx* KO mice by conventional homologous recombination using ES cells and we found that two independent KO mouse lines from two different ES clones showed the same phenotype. We believe these two independent lines indicate that the phenotype is caused by the deletion of the gene of interest. Finally, whether or not the KO mice showed the abnormal eye phenotype, we found that all mutants showed XCI impairments, indicating that *Ftx* could have been involved in proper regulation of XCI. We also suggest a relationship between *Xist* reductions and the severity of the variable phenotype of these KO mice. We believe that this is an important finding, not only for epigenetic research, but also in genetic and lncRNA research.

2. It is not clear which *Cre* mouse was used to generate *Ftx* or miRNA374/421 DKO mice.

Answer: We thank the reviewer for pointing this out. We used the CAG-flpe mouse to generate *Ftx* KO and the CAG-Cre mouse to generate miRNA374/421 DKO offspring. We have reported our method of generating *Ftx* KO mice previously (Soma et al. doi:10.1038/srep05181). Concerning miRNA374/421 DKO mice, we have corrected the information in the Methods section: line 387-388 in the revised manuscript.

3. Fig 1SD. *Ftx* expression in the eye samples should be examined in addition to the kidney samples that are reported in this figure. More importantly, *Ftx* expression in WT female mice samples needs to

be reported so one can evaluate the affect of Ftx heterozygous and homozygous deletion on Ftx expression.

Answer: Following the reviewer's comments, we have examined *Ftx* expression in the eye samples including WT, and heterozygous and homozygous KO samples, so we could evaluate the effect of *Ftx* expression in KO mice. As Reviewer 3 also requested information on *Xist* expression, we have summarized the expression data in our new Fig. S10, and they are mentioned in the Results section in lines 269-272. We have also added WT data to the new Fig. S1e.

4. Fig 2d-e. Southern blotting for female homozygous mutant miRNA374/421 mice is missing. More importantly in panel e, female WT levels for miRNAs 374 and 421 were not reported. Interestingly, in an earlier manuscript the authors show that male blastocysts lack miRNA374 and 421 expression, which is present in kidney samples from male mice. This data argue that Ftx and miRNA374/421 expression might be differently regulated in somatic cells. It is therefore essential to evaluate Ftx levels in eye samples from miRNA374/421 DKO mice.

Answer: We thank the reviewer for pointing this out. We had forgotten to show the Cre-mediated recombination of targeted locus in female heterozygous and homozygous mutants. Because the deleted fragment is quite small (less than 1 kb in size), we carried out PCR to check the Cre-recombination. We have included the results in the new Fig. 2e. Following the reviewer's suggestion, we also included female WT RT-qPCR results for miRNA374 and miRNA421 (see the new Fig. 2f), and *Ftx* RT-qPCR results for *miRNA374/421* DKO mice (see the new Fig. S1f, lines 133-136). As the reviewer pointed out, we agree that all transcripts including miRNA374 and miRNA421, and their host transcript *Ftx*, are differentially regulated in pre- and postimplantation stages. Imprinted expression of these transcripts at preimplantation stages disappeared consistently after implantation. We believe the important point of this manuscript is that *Ftx* disruption resulted in eye abnormalities, but miRNA DKO did not, suggesting that this lncRNA functions in eye development.

5. Fig 2f. I find inconsistentencies in the animal numbers presented in Figure 2f and those reported in Table S1 and S2, which will likely impact significance values presented in Fig 2f. Differences are as follows:

Table S1. Male *Ftx* consistentencies in the animal numbers presented in Figure *Ftx* +/-, n=118 (reported in Fig 2f) vs n=127 (Table S1).

Table S2. Female miRNA control, n=35 (reported in Fig2f) vs n=59 (Table S2). Female miRNA +/- KO, n=69 (reported in Fig 2f) vs n=45 (Table S2).

Answer: We thank the reviewer for pointing this out. We have corrected the numbers in the new Fig. 2g. In #64 ES line, male *Ftx* KO \bar{Y} : "1/103" (1.0%) has been corrected to "1/129" (0.8%), heterozygous *Ftx* female KO (+/-, -/+): "9/118" (7.6%) has been corrected to "9/127" (7.1%).

In Table S2, “+/+” in the top panel was corrected to “+/-“ (new Table S2), so the total numbers (n = 35, n = 69) in the old Fig. 2f have not been changed.

Although the total numbers of animals have been corrected, statistically significant differences were still detected between *Ftx*^{-/-} and miRNA KO animals, so there is no change to our argument.

6. Fig 3b. Animal numbers used to plot Fig 3b should be reported. More importantly, WT and mutant animal numbers vary significantly. Especially in a study like this where a stochastic phenotype is detected, I think it is important to compare littermates and I wonder if data reported in this figure were collected considering this fact.

Answer: Following the reviewer’s suggestion, we have included the embryo numbers in Fig. 3b. We argued that Fig. 3b shows statistically significant differences between male (-/Y) and female (-/-) *Ftx* KO mice. To make this easier to understand, we have included a horizontal line between the male and female mice. We understand it would be ideal to compare WT and ^{-/-} litter-mates. However, because *Ftx* is X-linked, it is impossible to do this. Instead we have analysed hundreds of eyes from a large number of embryos and carried out statistical analysis. The data showed a significant phenotypic difference between male KO and female KO mice. In addition, concerning the frequency of microphthalmia in C57/BL6N WT mice used in this study, Clea Japan Co. inform us that it is very rare: 0.44% in females (757/172431) vs. 0.04% in males (48/110913). This is consistent with our data shown in Figs 2g and 3b.

7. Fig 3d. How does the statics in Fig 3d correlate with Fig 3c data? There is inconsistency in the numbers especially for the % of female mice that present type 2 and 3 pathology.

Answer: We thank the reviewer for this comment. Our interpretation is that the reviewer is asking about the correlation between Figs 3b and 3d. The original Fig. 3b showed a summary of *all* examined samples, whereas Fig. 3d showed results obtained from a subset of WT and KO eyes classified by their appearance. Although dissection analysis was not carried out in all embryos, we think that the numbers were sufficient to conclude that the KO eyes classified as types 2 and 3 were smaller than in WT mice. To clarify this, we have added an explanation in the legend to Fig. 3d, lines 635-637, and also added the numbers of eyes examined in our new Fig. 3d.

8. Fig 4a. Were the mice used in microarray analyses, littermates? Microarray data was only supplemented as gene enrichment tables instead of individual gene expression changes. 4b. X-linked and autosomal gene expression changes should be presented as a scatter plot comparing wt and mutant samples. It is very difficult to interpret microarray results based on the short list provided for selected X linked genes in Fig 4d. The threshold applied on microarray data are at the border and might suggest no change. Detecting 1.2-1.7 fold change might purely a reflection of sample size. Applying more stringent cut offs would be necessary. If 2-fold change is applied, only 3 genes on X

show upregulation (considering this list is complete), which to this reviewer does not signify substantial XCI defect. These observations argue for a non-XCI related process in explaining the Ftx mutant mice pathology.

Answers:

For Fig. 4a:

We appreciate the reviewer's comment. Although we understand that studying litter-mates would be ideal, it was impossible to obtain WT and homozygous mutant embryos as litter-mates, because *Ftx* is an X-linked gene. Taking the possible variability of type 1 mutant eyes into consideration, we analysed 11 KO eyes (we duplicated 11 hybridization experiments) and carried out statistical analyses to evaluate the effects of KO. Following the reviewer's suggestions, all X-linked gene microarray data have been included in Table S8 and mentioned in lines 657-658.

Fig. 4b:

Following the reviewer's suggestion, we have added scatter plots comparing WT and mutant samples for presenting X-linked and autosomal gene expression changes as Fig. S2 (lines 183 in the revised manuscript).

Fig. 4d:

We think the threshold applied for microarray data was appropriate for the analysis. The maximum effect of gene expression level in impaired X-inactivation was twice as large as in WT mice. Thus, it is reasonable to judge a 1.2–2.0-fold change (FC) as reflecting impaired X-inactivation. Actually, three genes (*Ogt*, FC=1.57; *Tmem29*, FC=1.56; *Mecp2*, FC=1.55), listed in Fig. 4d, were confirmed to be upregulated in KO mice using RT–qPCR analysis, and these genes were shown to be misexpressed from the inactive X chromosome by RNA–FISH (Fig. 5) and RFLP analysis (newly added Figs S5–S7 in this revised MS). Furthermore, microarray control experiments supported this idea. In our control experiment, fold changes were calculated from WT females vs. WT males: 97% of X-linked genes were detected within $0.8 < FC < 1.2$, and five reported escapees were successfully detected as $FC > 1.2$. These data were consistent with the previous report on escapees (Berletch et al., PLoS Genet. 2015 Mar 18;11(3):e1005079. doi: 10.1371/journal.pgen.1005079). These results support the idea that the present threshold is valid. Although the reviewer was worried that detecting the 1.3–1.7 FC values might be purely a reflection of sample size, our data suggest that there was no significant size difference between WT and type 1 KO eyes at E13.5 (Fig. 3d, right panel). Taking these results together, we consider that our microarray analysis was valid for detecting XCI impairment.

9. Fig 4b. How does the authors explain X linked gene upregulation in male Ftx mutant mice?

Answer: The affected number of genes in male KO mice was noticeably smaller than in females (as shown in Fig. 4b, and the newly added Fig. S2). This suggests that *Ftx* has more significant functions in

females than in males. However, we cannot rule out possible roles in male mice and have added some comments in the Discussion (lines 319-323).

10. Fig 4e. Should be replaced with allele specific qRT-PCR analyses. Alternatively, Sanger Sequencing of gDNA and transcript analysis of cDNA at these loci should be done to validate allele specific transcription changes due to Ftx deletion.

Answer: We appreciate the reviewer's comment. In allele-specific RT-qPCR analysis, it is quite difficult to set the appropriate conditions for measuring gene expression correctly. Following the reviewer's comment, we carried out RFLP analysis to detect any misexpression from the inactive X chromosome in a KO-specific manner. We chose this method because it is more sensitive than Sanger sequencing. We confirmed that ectopic expression from the inactive X was detected in *Ftx* KO mice but not in WT mice, supporting the idea that *Ftx* KO showed XCI impairment. These results are summarized in the new Figs S5-S7, and comments have been added in lines 223-240 in the revised manuscript.

11. Fig 5. Total cell numbers analysed, and the cell numbers that present mono- or biallelic expression for 3 X linked genes were not reported. The authors indicate that they can detect Xist RNA cloud in 85% of cells. I wonder if this was factored in in their biallelic RNA-FISH analyses in panel b or c.

Answer: Following the reviewer's suggestion, we have added the total cell numbers and the percentages showing mono- or biallelic expression for three genes. Because Reviewer 1 also had the same concern, we have included the number of cells with no *Xist* cloud signal, and redrawn the bar graph (new Figs 5b and 5c).

12. Fig 5a-b-c. I am confused with the results reported here. For example, cells that are categorized as light pink are supposed to show expression of X-linked genes that only express from Xi. If this data is correct, does it mean Ftx cause loss of active X chr specific expression of these genes as well? This reviewer thinks this pattern might be reflective of those cells in which RNA-FISH did not work efficiently. It is also important to show that the second RNA signal detected for X-linked genes is indeed coming from the second X chr copy. Were wt and mutant eye samples collected from littermates? Have the authors examined karyotype of these mice? Were there any chromosomal abnormalities?

Answer: In general, RNA-FISH cannot detect a target signal in 100% of cells. Even in the easiest cases, such as detecting the *Xist* cloud, the signal was detected in 85% of all cells. As shown in the new Fig. S3a, three X-linked genes were detected in 40% of male cells, which were used as our controls for RNA-FISH. We think that the sensitivity limitation of this system is generally accepted. Thus, light pink is intended to show expression of X-linked genes from the inactive X, but it does not necessarily mean that this gene was not expressed from the active X. The most important point was that dark pink cells were actually

detected in KO mice, but barely detected in WT mice. This difference was statistically significant for all genes examined in Figs 5b and 5c, and it was consistent with other results: DNA microarray, RT-qPCR, and the newly added RFLP analysis that showed impaired XCI in a KO-specific manner. The two spot signals which are very close to each other are thought to represent early replicated X chromosomes. To avoid misunderstanding this pattern as aneuploidy, we combined grey with black cells (in the old version), as in the new Figs 5b and 5c in the revised manuscript.

13. *Xist* expression looks intact in FISH analyses (Fig 5a). However, the authors show by RT-PCR that there is ~30% decrease in *Xist* transcript. This reviewer thinks that if the authors would like to suggest that this is causal to upregulation of X-linked genes, it is necessary to determine changes in heterochromatin marks associated with these loci and perform allele-specific RNA-Seq.

Answer: Because Reviewer 1 also had the same concern about a discrepancy between *Xist* RT-qPCR and RNA-FISH, we have quantified the *Xist* cloud area in KO (*Ftx*^{-/-}) and compared it with that of WT cells. The size of the *Xist* cloud in KO cells was slightly—but significantly—smaller than in the WT cells. We have eliminated line 219 (old version) “the clouds were morphologically indistinguishable between these cells”, and have added comments in lines 219-222 and 245-246 and in the legend for new Fig. S4.

14. *Based on these comments, this reviewer thinks that the XCI phenotype in female *Ftx* mutant mice is very mild to none and the data presented is not sufficient to support the working model of the authors.*

Answer: We agree that *Ftx* KO mice showed a milder phenotype than has been reported previously for XCI mutant mice. In this revision, we have added allelic expression analysis (Figs S5–S7), which shows ectopic expression from the Xi in KO mice. Combined with the previous data including DNA microarray, qPCR, and RNA-FISH analyses, we conclude that there was apparent XCI impairment, but not as severe as in previously reported mutants. Because these KO mice were born alive and fertile, we think this is a new category of XCI mutant mouse and might serve as a novel model to explain female-specific human genetic diseases that show an unusual X-linked inheritance pattern. Following the reviewer’s comment, we have removed our working model in Fig. 6f (old version). We would like to carry out a detailed analysis of XCI mechanisms in another paper in the future.

Reviewer #3 (Remarks to the Author):

This manuscript by Hosoi et al., studies the effects of loss of *Ftx* in vivo. *Ftx* heterozygous and homozygous knockout female mice display an eye phenotype that resembles microphthalmia. This phenotype is not fully penetrant and only present in female offspring suggesting a relation to X chromosome inactivation. The authors indeed find reduced expression of *Xist* and concomitant

up-regulation of X linked genes in mice with an eye phenotype. In addition, genes involved in eye development appear downregulated in affected females, suggesting a link between loss of Xist expression and reduced expression of genes involved in eye development. The findings are in line with previous reports on conditional Xist knockout mice showing various phenotypes. This study indicates that Ftx mutations appear to affect eye development more than other tissues and organs, suggesting a more important role for Xist expression in this tissue. I have several suggestions to improve the manuscript.

Answer: We appreciate the reviewer's positive view and helpful suggestion about a report that *Xist* conditional KO mice showed variable phenotypes. We have added this information to the Discussion, line 334-335. To make the impairment status of random XCI in KO mice clearer, we have performed additional experiments, which yielded the following results.

- 1) RFLP analysis at E13.5 revealed ectopic expression of X-linked genes from the inactive X, suggesting that *Ftx* is involved in random XCI and acts in cis.
- 2) RNA-FISH at E6.5 showed that random XCI was impaired in its early stages in *Ftx* KO embryos, suggesting its potential role in the initiation of random XCI.
- 3) Allelic expression analysis of X-linked genes at E9.5 showed that deleting *Ftx* on one X chromosome did not skew XCI toward the WT X chromosome.
- 4) Quantification of *Xist* expression in type 2 brains supported the idea that there could be a relationship between *Xist* reduction and the severity of the variable phenotype seen in our KO mice.

These results are summarized in our new Figs 6f and S4-S10 and are mentioned in the Results and Discussion (lines 223-240, 249-255, 255-262 and 289-296). We hope the revised manuscript will satisfy the reviewer's concerns.

One major question that is not addressed by the authors is what is really going wrong. Is X chromosome inactivation initially random and initiated normally followed by loss of X chromosome inactivation, or is initiation of X chromosome inactivation affected? To address this question studies on early embryo's need to be performed to test whether loss of XCI is present from the start with indicates that initiation is affected.

Answer: Because Reviewer 1 also made the same comment, we have carried out RNA-FISH analysis of *Ftx*^{-/-} embryos at E6.5 to examine whether the initiation of random XCI was affected in KO mice. As a result, impairment of XCI was observed at its initiation. These results are summarized in the new Fig. S8, and mentioned in the Results, lines 249-255.

In this respect it will be important to test whether XCI is random, for instance by generating F1 B6: mollosinus mice. I think this is important to know, because the phenotype might be hard to explain if X chromosome inactivation will be skewed towards inactivation of the wild type X chromosome.

Answer: To examine whether random XCI was affected in *Ftx* heterozygous KO mice, we have carried out RFLP analysis of six X-linked genes in (B6 × JF1) F1 E9.5 embryos. As shown in the revision (lines 255-262, and the new Fig. S9), deleting *Ftx* on one X chromosome did not skew XCI toward the WT X chromosome in female somatic cells. No obvious skewing effect on random XCI was observed.

Second the Ftx knockout allele was generated in a 129Sv background and was then backcrossed into the B6 background. As the Ftx mutation is linked to the X inactivation center, and the 129/Sv and B6 strains retain different Xce strengths it is expected that Ftx^{+/-} heterozygous mice might display skewed XCI which may affect the phenotype.

Answer: We apologize for the confusing description. *Ftx* knockout mice were generated using 100% pure B6 background ES cells and maintained in a B6N line (Soma et al. 2014 Sci. Reports, ref. 16). Therefore, there was no need to consider the effect of genetic background when we examined random XCI in *Ftx* KO mice. By contrast, miRNA DKO mice were generated in a 129Sv background ES cell line and then backcrossed into the B6 strain (this manuscript). We have described the genetic background of *Ftx* KO in lines 342 and 367–370 (old version), and in lines 347-348 and 376 of the new version as well.

In the final paragraph the effects of Ftx homozygous mutations on Xist and X linked and autosomal gene expression are studied, showing reduced Xist expression and concomitant effects on both X linked and autosomal genes. Here I would have liked to see the same effects in the affected heterozygous mice.

Answer: To examine the effect of heterozygous *Ftx* KO, we have examined (1) the allelic expression of affected X-linked genes (*Ogt*, *Mecp2*) in heterozygous ^{+/-} KO eyes using RFLP analysis; (2) quantified *Xist* expression level in these eyes, and compared it with those of WT, and homozygous ^{-/-} KO mice. As shown in the revision (lines 223-240 and 269-272, new Figs S5–S7, and new Fig. S10), heterozygous KO mice also showed impaired XCI and diminished *Xist* expression.

In the discussion it is concluded that ‘So far, there have been no reports of mutantother than embryonic death’. This is not completely true as the Lee lab also deleted Xist specifically in the haematopoietic lineage resulting in loss of XCI and anaemia.

Answer: Following the reviewer’s comment, we have eliminated line 384 and the wording has been changed for clarity (lines 332-338 in the revision).

Is there an overlap in the limited subset of X linked genes that are affected in Ftx^{-/-} homozygous knockout mice?

Answer: We think that the listed genes in Fig. 4d are overlapping genes that tended to be commonly affected in all 11 *Ftx*^{-/-} eyes examined, because the list was calculated from the average of the values of 11 eyes (i.e., 22 hybridization results).

Figure 2e, why was Ftx expression not examined?

Answer: We have carried out RT-qPCR analysis of *Ftx* in miRNA DKO mice as shown in Fig. S1f. *Ftx* expression was not affected in miRNA DKO mice, see lines 134-136.

Where are Ftx and Xist in Figure 4C, these genes need to be indicated.

Answer: We have added the *Xist* location in Fig. 4c. Because the signal intensity of *Ftx* was lower than 500, this image does not include the *Ftx* spot. We have added comments in the legend for Fig. 4c, lines 654-655.

Figure 6a, I suppose this is data obtained on RNA isolated from the eye.

Answer: We thank the reviewer for pointing this out. The data were obtained from the DNA microarray experiments. We have added “the signal intensity of the microarray” on the Y-axis in Fig. 6a for clarification.

Reviewers' comments:

Reviewer #1 (Remarks to the Author):

In the revised manuscript, Kobayashi and colleagues have provided a significant number of additional data, which answered most of my concerns.

In addition, the text has been clarified and the different hypothesis better discussed.

The final manuscript will be of interest for researchers in X-chromosome inactivation and mouse development.

Reviewer #2 (Remarks to the Author):

I thank Hosoi et al. for addressing majority of my concerns. I have the following major comments that I believe will help in improving the manuscript further.

Lane 85. "Such misexpressed genes were distributed along the X chromosome, indicating that Ftx could be involved in inactivating the entire region of the X chromosome". I do not think the authors provided any evidence that Ftx provides inactivation on an entire region, or any region for that matter, of the X chromosome. There are 3 genes that show upregulation within the $FC > 2.0$ and the rest of the genes which were shown to be within $2.0 > FC > 1.2$ are positioned across the X chromosome. Thus, the phrase should be changed accordingly.

Fig 5. The authors reported that 3.5-8.7% of the Ftx mutant eye cells show expression of Tmem29, Mecp2, Ogt only from the inactive X chromosome (Xi). It was noted in the rebuttal that this might be due to inefficiency of the RNA FISH. Based on this argument, we can apply the same logic for the other subpopulations represented in the graph. It is important that the RNA FISH images of cells in which only Xi specific transcription are observed for overexpressing genes, are included in the manuscript as their % is similar to what is observed for biallelic gene expression. I would like to also suggest that the authors perform RNA FISH for Xist and for one or two X-linked transcripts that do not show upregulation. This way one can evaluate the sensitivity of their RNA FISH and XCI pattern in eye cells.

Table S8 and Fig 4b. The authors included the excel table containing X linked genes, but if I understand correctly there is discrepancy in the number of genes plotted in Fig 4b and Table S8. Number of genes that show up and down regulation should be reexamined and represented correctly in Fig 4b. Moreover, since the authors evaluated autosomal gene expression in a graph, a separate table showing changes in autosomal gene expression in WT vs Ftx mutants should be included in the manuscript. This data would be complementary to Fig 6e and Fig S12.

Table S8 and Fig 6a. Authors plotted Xist signal intensity based on the microarray data in Fig 6a. Examining Table S8 for Xist expression, one detects 7 different Xist probes represented on the microarray. None of the intensity values detected for these probes match to the numbers that are plotted as Xist intensity in Fig 6a. I am not sure whether the authors plotted Fig 6a based on the average of Xist signal intensity or raw values. Thus, this figure requires attention and correct numbers should be plotted in the Fig.

Fig S8. The authors noted in the rebuttal that they examined if Ftx is important for the XCI initiation or maintenance by examining expression of X-linked genes in E6.5 (XCI initiation stage) and in E13.5 (XCI maintenance). They indicated that they evaluated both E6.5 and E13.5 for Mecp2 expression in WT and Ftx mutant embryos and detected 2.7% and 9.7% cells showing misexpression for Mecp1 and

that they presented the data in Fig S8. This data was presented incomplete and I have concerns about the conclusions drawn. First, only E6.5 data is presented and E13.5 data is missing in Fig S8. Second, close examination of the WT Xist FISH image shows that only ~10% of the cells express Xist (n=3 embryos). However, in the graph (Fig S8b) this number was reported as 89-90%. This might be because the image resolution is low, or the images shown are not representative of what is plotted in the graph. It is more likely that there were issues in detecting Xist and Mecp2 RNA signals properly in this experiment. Moreover, the authors reported 1.7% cells showing biallelic expression for Mecp2 in WT embryos while Ftx mutant embryos show only 1% of cells with biallelic Mecp2 expression. This means that 2.7% of cells show skewed (only Xi specific) expression and I do not see a close up RNA FISH image showing a representative cell for this population. Based on these data, I do not think one can conclude that Ftx has a role during XCI initiation or maintenance. The experiments need to be repeated using E6.5 and E13.5 embryos with proper controls showing RNA FISH signal can be detected in 80-90% of the cells.

Reviewer #3 (Remarks to the Author):

This manuscript by Hosoi et al has significantly improved, and most of my questions have been addressed properly. I have two questions remaining that the authors might want to discuss in more detail. If E6.5 embryos appear to initiate XCI nearly as good as WT, initiation might be less affected, but it more looks like the Xi is improperly installed or maintained due to low levels of Xist either during the initiation or establishment and maintenance phases. Second, why is XCI skewed in Ftx +/- mutant cells in vitro and not in vivo, are the alleles different?

We would like to thank the reviewers for their comments. We have revised our manuscript accordingly. The new text is highlighted in yellow in the revised manuscript.

Reviewers' comments:

Reviewer #1 (Remarks to the Author):

In the revised manuscript, Kobayashi and colleagues have provided a significant number of additional data, which answered most of my concerns.

In addition, the text has been clarified and the different hypothesis better discussed.

The final manuscript will be of interest for researchers in X-chromosome inactivation and mouse development.

Answer:

We very much appreciate the reviewer's comment and are pleased to see that he/she was satisfied with the revised manuscript.

Reviewer #2 (Remarks to the Author):

I thank H. et al. for addressing majority of my concerns. I have the following major comments that I believe will help in improving the manuscript further.

Lane 85. "Such misexpressed genes were distributed along the X chromosome, indicating that Ftx could be involved in inactivating the entire region of the X chromosome". I do not think the authors provided any evidence that Ftx provides inactivation on an entire region, or any region for that matter, of the X chromosome. There are 3 genes that show upregulation within the $FC > 2.0$ and the rest of the genes which were shown to be within $2.0 > FC > 1.2$ are positioned across the X chromosome. Thus, the phrase should be changed accordingly.

Answer:

We appreciate the reviewer's comment. Following the referee's comment, we have changed the wording in lines 86-88.

Fig 5. The authors reported that 3.5-8.7% of the Ftx mutant eye cells show expression of Tmem29, Mecp2, Ogt only from the inactive X chromosome (Xi). It was noted in

the rebuttal that this might be due to inefficiency of the RNA FISH. Based on this argument, we can apply the same logic for the other subpopulations represented in the graph. It is important that the RNA FISH images of cells in which only Xi specific transcription are observed for overexpressing genes, are included in the manuscript as their % is similar to what is observed for biallelic gene expression. I would like to also suggest that the authors perform RNA FISH for Xist and for one or two X-linked transcripts that do not show upregulation. This way one can evaluate the sensitivity of their RNA FISH and XCI pattern in eye cells.

Answer:

Following the reviewer's comments, we have added RNA-FISH images of cells in which only Xi-specific transcriptions were observed for overexpressed genes (indicated as light pink cells in the graphs in Fig. 5b, c), to a new Fig. 5a. In the previous revision, RNA-FISH results for the non-upregulated gene *Pgk1* was added to Fig. S3b. The results confirmed that not all X-linked genes were affected, and that the RNA-FISH experiment worked well.

Table S8 and Fig 4b. The authors included the excel table containing X linked genes, but if I understand correctly there is discrepancy in the number of genes plotted in Fig 4b and Table S8. Number of genes that show up and down regulation should be reexamined and represented correctly in Fig 4b. Moreover, since the authors evaluated autosomal gene expression in a graph, a separate table showing changes in autosomal gene expression in WT vs Ftx mutants should be included in the manuscript. This data would be complementary to Fig 6e and Fig S12.

The graph in Fig. 4b shows the number of genes that are significantly differentially expressed between WT and KO eyes. Therefore, we plotted the number of up- and downregulated genes ($P < 0.05$). To clarify this, we have added P values (< 0.05) in line 650 in the revision. Following the reviewer's comments, we have provided a list of all the autosomal genes as a new Supplementary Table S9 and have added comments in lines 660-662. We have also submitted all our microarray data to NCBI (GEO accession number: GSE100700).

Table S8 and Fig 6a. Authors plotted Xist signal intensity based on the microarray data in Fig 6a. Examining Table S8 for Xist expression, one detects 7 different Xist probes represented on the microarray. None of the intensity values detected for these probes match to the numbers that are plotted as Xist intensity in Fig 6a. I am not sure whether the authors plotted Fig 6a based on the average of Xist signal intensity or

raw values. Thus, this figure requires attention and correct numbers should be plotted in the Fig.

Answer: In Fig. 6a, we used the value of probe “A_30_P01022001”. Please note that Table S8 contains two extra columns: “WT Female average” and “KO Female average”, which are not included in Fig 6a graph. All seven Xist probes showed similar expression patterns, and the average value of these probes showed significant reduction of Xist expression in KO eyes ($P = 0.015$). Therefore, we selected a representative data sample: “A_30_P01022001” for Fig. 6a. For the reviewer’s information, we have attached a graph showing the average values of these seven probes (see the figure below): it is not included in the revised manuscript. We have added this information in lines 684–687 for clarity.

Fig. Average values for all seven Xist probes

Fig S8. The authors noted in the rebuttal that they examined if Ftx is important for the XCI initiation or maintenance by examining expression of X-linked genes in E6.5 (XCI initiation stage) and in E13.5 (XCI maintenance). They indicated that they evaluated both E6.5 and E13.5 for Mecp2 expression in WT and Ftx mutant embryos and detected 2.7% and 9.7% cells showing misexpression for Mecp1 and that they presented the data in Fig S8. This data was presented incomplete and I have concerns about the conclusions drawn. First, only E6.5 data is presented and E13.5 data is missing in Fig S8. Second, close examination of the WT Xist FISH image shows that only ~10% of the cells express Xist (n=3 embryos). However, in the graph (Fig S8b) this number was reported as 89-90%. This might be because the image resolution is low, or the images shown are not representative of what is plotted in the graph. It is more likely that there were issues in detecting Xist and Mecp2 RNA signals properly in this experiment. Moreover, the authors reported 1.7% cells showing biallelic expression for Mecp2 in WT embryos while Ftx mutant embryos show only 1% of cells with biallelic Mecp2 expression. This means that 2.7% of cells show skewed

(only Xi specific) expression and I do not see a close up RNA FISH image showing a representative cell for this population. Based on these data, I do not think one can conclude that Ftx has a role during XCI initiation or maintenance. The experiments need to be repeated using E6.5 and E13.5 embryos with proper controls showing RNA FISH signal can be detected in 80-90% of the cells.

First, only E6.5 data is presented and E13.5 data is missing in Fig S8.

Answer:

To avoid misunderstanding, we have changed the wording in lines 256-257 to clarify the positions of RNA-FISH results for E6.5 (Supplementary Fig. S8b) and E13.5 (Fig. 5b, c) in the main text.

Second, close examination of the WT Xist FISH image shows that only ~10% of the cells express Xist (n=3 embryos). However, in the graph (Fig S8b) this number was reported as 89-90%.

Answer:

Because we aimed to analyse the epiblast cells at E6.5 to evaluate the effects of *Ftx* deletion on random XCI, we observed E6.5 samples using whole-mount 3D RNA-FISH combined with anti-Oct-3/4 immunofluorescence to discriminate epiblast from endoderm cells. Optical sections of whole embryos were obtained using confocal microscopy Z-stacks and were used to reconstruct 3D confocal images. Supplementary Fig. S8 shows only one section of the Z-stack images, so one might expect to find only ~10% of the cells expressing *Xist*. We estimated the total cell counts using reconstructed 3D imaging. The results summarized in Supplementary Fig. S8b show that the *Xist* RNA-FISH signal was detected in 90% of the cells. Thus, there is no discrepancy. The technique is described in detail in the Methods section. To avoid misunderstanding, we have also added further brief methods information in the legend for Supplementary Fig. S8. We have also attached a movie of WT control embryos in response, solely for the reviewers; it is not to be included in the revised manuscript. We believe this movie will help the reviewers understand how the *Xist* signals were detected and counted in the 3D reconstructed images.

This means that 2.7% of cells show skewed (only Xi specific) expression and I do not see a close up RNA FISH image showing a representative cell for this population.

Answer:

We observed the E6.5 sample using whole-mount 3D RNA-FISH. As described above, Fig. S8 shows only one section of Z-stack images, so one might not find cells showing skewed (only Xi specific) expression. We believe there are no ambiguities in our method.

Reviewer #3 (Remarks to the Author):

This manuscript by H. et al has significantly improved, and most of my questions have been addressed properly. I have two questions remaining that the authors might want to discuss in more detail. If E6.5 embryo's appear to initiate XCI nearly as good as WT, initiation might be less affected, but it more looks like the Xi is improperly installed or maintained due to low levels of Xist either during the initiation or establishment and maintenance phases. Second, why is XCI skewed in Ftx^{+/-} mutant cells in vitro and not in vivo, are the alleles different?

Answer: We very much appreciate the reviewer's comment and are pleased to see that he/she was satisfied with the revised manuscript. We are also interested in the questions raised by the reviewers and would like to study these in detail in another paper.

REVIEWERS' COMMENTS:

Reviewer #2 (Remarks to the Author):

I thank Kobayashi and colleagues for addressing my concerns and comments by improving the manuscript, clarifying the data and text. I believe the final manuscript is timely and will provide interesting insights to the role of Ftx on the X-chromosome inactivation and its link to mouse development.